# Neural evidence for lexical parafoveal processing

Yali Pan [1,2✉], Steven Frisson[1,2] & Ole Jensen [1,2]

In spite of the reduced visual acuity, parafoveal information plays an important role in natural reading. However, competing models on reading disagree on whether words are previewed parafoveally at the lexical level. We find neural evidence for lexical parafoveal processing by combining a rapid invisible frequency tagging (RIFT) approach with magnetoencephalography (MEG) and eye-tracking. In a silent reading task, target words are tagged (flickered) subliminally at 60 Hz. The tagging responses measured when fixating on the pre-target word reflect parafoveal processing of the target word. We observe stronger tagging responses during pre-target fixations when followed by low compared with high lexical frequency targets. Moreover, this lexical parafoveal processing is associated with individual reading speed. Our findings suggest that reading unfolds in the fovea and parafovea simultaneously to support fluent reading.

[1] Centre for Human Brain Health, University of Birmingham, Birmingham, UK. [2] School of Psychology, University of Birmingham, Birmingham, UK.
✉email: yalipan666@gmail.com

Humans have developed the remarkable skill of reading, allowing for efficient acquisition of information from busy pages or screens of text. Given the importance of written text for communication, individuals with reading disabilities are highly disadvantaged in modern society. Yet, we know little about the neuronal mechanism underlying natural reading.

It is well known that reading is severely impaired when masking out the parafoveal area (i.e., 2–5 visual degrees to the current fixation)[1–5]. This finding shows that parafoveal information plays a critical role in fluent reading regardless of its relatively low visual acuity. How much and what type of information is previewed from the parafoveal area is highly controversial for eye movement control models[6]. Lexical information which is related to word frequency (i.e., how often a given word occurs in the language) is important for word recognition and impacts how we move our eyes (for a review see ref. [7]). Serial attention shift models maintain that lexical processing is restricted to one word at a time[8–13], but that attention can be shifted to the next word before the eyes do, allowing significant parafoveal processing[14]. According to the mechanism described in the E-Z Reader model, the parafoveal processing can explain, for example, word skipping effects[15]. In contrast, parallel graded processing models assume that attention is allocated to several words within a reader's perceptual span in a graded way[16,17] (for a recent model see the OB1- reader[18]). According to this framework, lexical information of both foveal and parafoveal words is extracted in parallel.

Most studies based on eye-tracking have produced data in support of serial attention shift models. This is based on the finding that fixation durations on a given foveated word is not impacted by the lexical frequency of the upcoming parafoveal word, the parafoveal-on-foveal effect[19–21]. However, while eye-tracking studies have been hugely informative, the technique only indirectly measures parafoveal processing. For instance, a few studies applied the gaze-contingent boundary paradigm and found a delayed parafoveal-on-foveal effect, where the fixation durations for word $n + 1$ were modulated by the preview difficulty of word $n + 2$[22–24]. Here, we applied a technique, rapid invisible frequency tagging (RIFT)[25] in combination with magnetoencephalography (MEG) to measure parafoveal processing, i.e., attention allocated to upcoming words in the parafovea. RIFT measures the neuronal excitability associated with attention by flickering stimuli at high frequencies that are invisible to participants. In previous MEG studies, we have demonstrated that RIFT captures covert attention, reflected by stronger tagging responses for attended compared with unattended stimuli[25–28].

In this work we aimed at answering if lexical information is retrieved from upcoming words in the parafovea during natural reading. We flickered the parafoveal (target) words at 60 Hz and measured the tagging responses during the current (pre-target) fixation. If the tagging responses during pre-target fixations are modulated by the lexical information of the target words, then it would indicate that lexical information can be extracted from the parafovea and provide neural evidence consistent with parallel models.

## Results

**No lexical parafoveal processing effect on eye movement data.** In the present study, 39 participants read 228 sentences in total (composed of two sets of sentences). All sentences were plausible and contained unpredictable target words of either low or high lexical frequency (see Supplementary Methods for plausibility and predictability pre-tests details). Word length for both pre-target and target words were matched with respect to low and high lexical frequency of the target words (Table 1). Target words were flickered at 60 Hz throughout the reading of each sentence while the neuronal activity was measured by MEG (Fig. 1). When participants fixated on the pre-target word, the flickering target induced reliable tagging responses at 60 Hz, reflecting the neural resources associated with parafoveal processing. Thus, this experimental design allowed us to investigate neural activity associated with the processing of the next word without interfering with natural reading.

In line with previous studies[19–21,23,29], we did not find an effect of target word lexical frequency on pre-target first fixation durations (i.e., the duration of the first fixation on a word) ($t_{(38)} = 0.17$, $p = 0.86$, $d = 0.03$, two-tailed pairwise $t$-test, Fig. 2). This finding demonstrates that parafoveal lexical processing was not reflected in the eye movement data. However, the target fixation durations were longer for low compared with high lexical frequency targets ($t_{(38)} = 6.94$, $p = 3 \times 10^{-8}$, $d = 1.11$, two-tailed pairwise $t$-test). This classic word frequency effect indicates a successful manipulation of target lexical frequency. We observed the same pattern using gaze durations (i.e., the sum of all fixations on a word when it is first encountered during reading; Supplementary Fig. 1).

**Rapid invisible frequency tagging captures lexical parafoveal processing.** We analyzed the MEG data to uncover the brain activity associated with lexical processing before saccading to the target word. A measure of time-resolved coherence between the 60 Hz visual flicker and the brain activity was used to quantify the tagging responses (see "Methods" section for details). Since coherence is based on the phase relationship between the photic driving signal and the brain response, it is a more sensitive measure than power of the brain response at the tagging frequency. In addition, coherence is quite stable even at a high-frequency band[30].

Since mainly sensors over visual areas responded to the visual flicker, we first identified the sensors with robust tagging responses. We compared the 60 Hz visual flicker-to-MEG coherence during pre-target fixations (caused by the target flickering in the parafovea) with a baseline period (the cross-fixation presented before sentence onset). Robust tagging responses were found over the left visual cortex sensors (Fig. 3a), reflecting the neural resources associated with parafoveal

**Table 1 Characteristics of words used in the current study.**

|  | Pre-target | Low frequency target | High frequency target | Post-target |
|---|---|---|---|---|
| Word frequency | 359.9 (1109.3) | 5.3 (4.5) | 95.3 (135.5) | 569.3 (1734.7) |
| Word length | 6.1 (1.5) | 5.8 (0.8) | 5.8 (0.8) | 6.7 (1.7) |
| Position | 5.7 (2.3) | 6.7 (2.3) | 6.7 (2.3) | 7.7 (2.3) |

All values here are mean values, standard deviations are in parentheses. Low (<10) and high lexical frequency (>30) target words are reported in terms of the total CELEX frequency per million[72]. Word length is the number of letters in a word. Position refers to the location in a sentence where the target word is presented and is measured in the number of words. The sentences were 11.2 ± 2.1 words long. Source data are provided as a Source Data file.

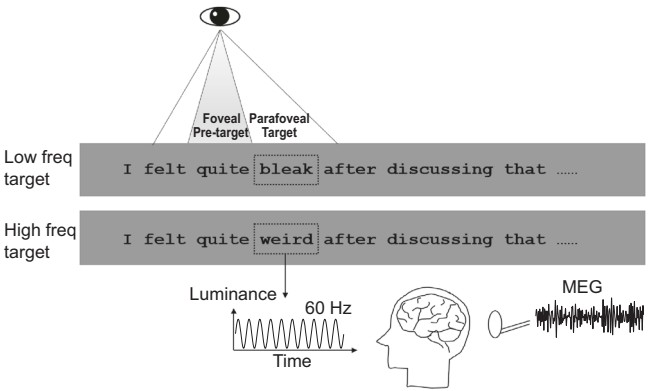

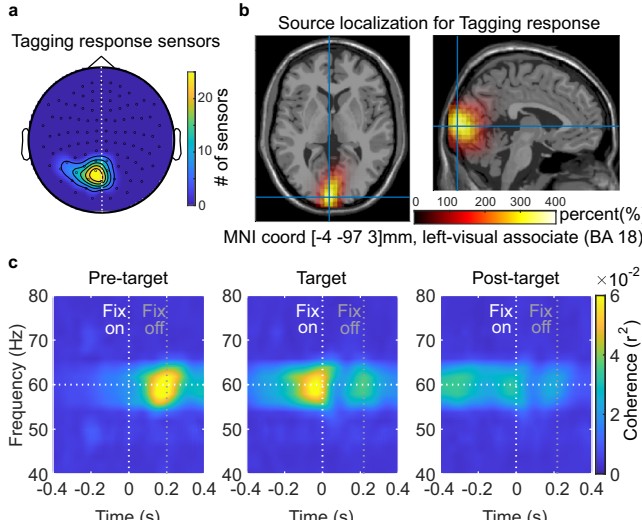

**Fig. 3 Neural responses of the rapid invisible frequency tagging. a** Topography for sensors from all participants that showed stronger tagging responses during the pre-target period (flicker) compared with the baseline period (no-flicker, n = 26). **b** These tagging responses were localized in the left visual cortex. **c** The time-course of neural tagging responses during the pre-target (left panel), target (middle panel), and post-target fixation periods (right panel). Vertical white and gray lines indicate fixations onsets and average offsets.

**Fig. 1 The reading task.** Participants (n = 39) read sentences silently, while eye-movements and brain activity were recorded. Each sentence contained either a low or high lexical frequency target word (see dashed rectangle; not shown in the experiment). A Gaussian smoothed patch beneath the target word was flickered at 60 Hz continuously when the sentence was on the screen. This allowed us to measure neural responses associated with lexical parafoveal using rapid invisible frequency tagging (RIFT). One-quarter of the sentences were followed by a simple yes-or-no comprehension question to ensure that participants read the sentences carefully. Freq frequency.

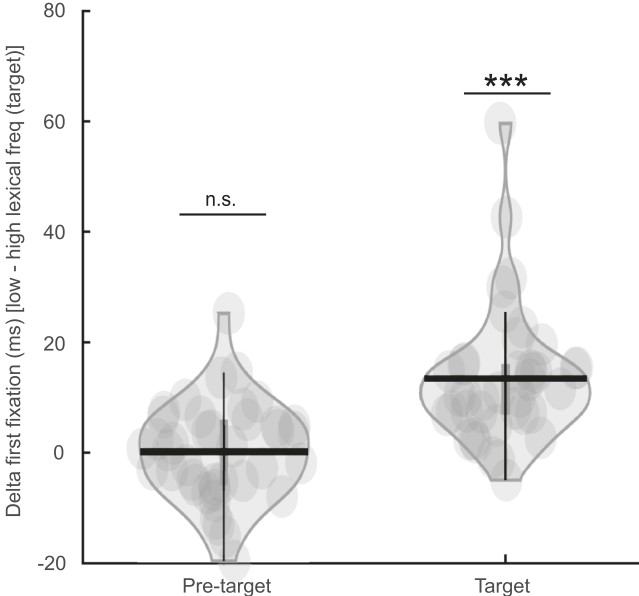

**Fig. 2 Eye movement metrics.** The first fixation duration difference for pre-target and target words when comparing low versus high lexical frequency target words. (***p = 2.97 × 10e−8, n = 39, two-sided pairwise t-test). The horizontal bar in the violin plots indicates the mean value; each dot represents one participant. Source data are provided as a Source Data file.

processing. This analysis was conducted by pooling data over both target lexical frequency conditions (for details see "Methods" section). In 26 out of the 39 participants, one or more sensors showed significant tagging responses (Fig. 3a, 5.4 ± 4.0 sensors per participant, mean ± SD; for topography of each participant, see Supplementary Fig. 2). The subsequent analyses were based on these participants and sensors. We also did the same tagging sensor selection procedure but used 60 Hz power instead, which was less sensitive than 60 Hz coherence (Supplementary Fig. 3).

A source modeling approach revealed that the generators of this 60 Hz coherence were localized in the early visual cortex (Brodmann area 17, 18; Fig. 3b). We also compared the difference

in sources of frequency tagging for low-frequency and high-frequency targets using the same source modeling approach; however the signal difference was not strong enough to produce a reliable effect. The time course of the 60 Hz coherence is shown in Fig. 3c. Note the robust 60 Hz coherence from the target word when fixating on the pre-target word (see Supplementary Fig. 4 for the power spectrum of pre-target intervals, with peaks around 60 Hz). These results demonstrate that RIFT is a sensitive tool for measuring brain activity associated with parafoveal processing during natural reading.

**Neural evidence for lexical parafoveal processing.** Next, we addressed if the RIFT responses were modulated by the lexical frequency of the target words. Our key finding was that the coherence at 60 Hz during the pre-target fixation was stronger when followed by a low compared with a high lexical frequency target word (Fig. 4a). To ensure that the coherence in the pre-target fixation intervals was not contaminated by temporal smoothing from the target fixation, the time window for averaging coherence was adjusted individually according to the shortest pre-target fixation duration. For each participant, we identified the shortest duration (t) over all pre-target fixations (t was 88.3 ± 8.9 ms across participants, mean ± SD). Next, pre-target coherence for the two conditions was averaged within this time window, then averaged over the sensors of interest to obtain the grand average. Because the number of trials biases the magnitude of the coherence measure, we subsampled the same number of trials for both conditions in each participant (by randomly selecting trials from the condition with more trials). The difference in coherence across participants was compared using a paired t-test demonstrating that the flicker response was significantly stronger for target words in the parafovea with a low compared to high lexical frequency (Fig. 4b, $t_{(25)} = 2.20$, p = 0.037, d = 0.43, two-tailed pairwise t-test). In sum, we found neural evidence in support of lexical information being extracted from the parafovea. Moreover, this lexical parafoveal processing also modulated pre-target coherence onset latency when both

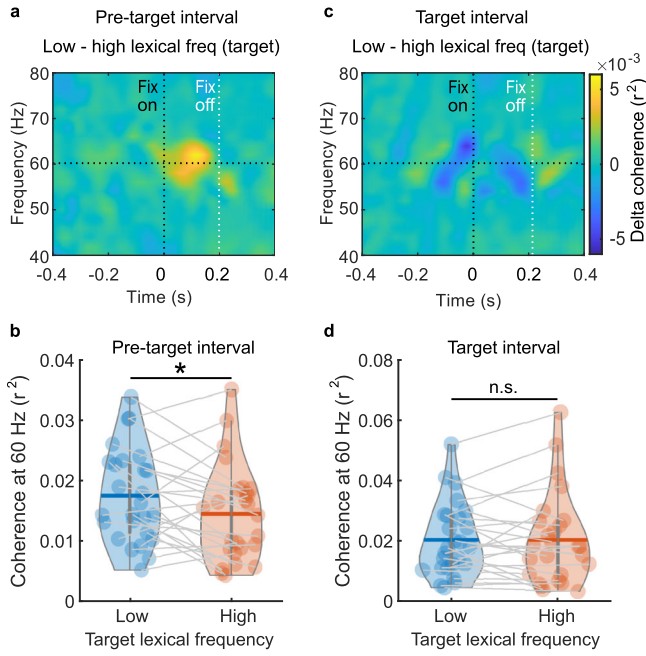

**Fig. 4 Neuronal evidence for lexical parafoveal processing. a** Time-resolved coherence differences during pre-target word fixations (low minus high lexical frequency target words; $n = 26$) revealing a lexical parafoveal effect around 100 ms after fixation onset. **b** The averaged pre-target coherence at 60 Hz during pre-target fixations for low (blue) and high (orange) lexical frequency target words (*$p = 0.037$, $n = 26$, two-sided pairwise $t$-test). **c**, **d** During target word fixations we did not observe a coherence difference with respect to lexical frequency ($p = 0.992$, $n = 26$, two-sided pairwise $t$-test). Horizontal bars in the violin plots indicate mean value. Each dot presents one participant. Source data of panel **b** and **d** are provided as a Source Data file.

pre-target and target words were short (see Supplementary Fig. 5 and Supplementary notes).

**No RIFT contamination from the foveal processing.** Next we tested whether the lexical frequency effect could also be observed during target fixations (Fig. 4c). As in the pre-target analysis, the averaging time window for the coherence was the minimum target fixation duration for each participant (87.1 ± 9.4 ms, mean ± SD). We observed no significant coherence difference when fixating on low versus high lexical frequency target words (Fig. 4d; $t_{(25)} = 0.01$, $p = 0.992$, $d = 0.002$, two-tailed pairwise $t$-test). We conclude that the lexical effect that we observed during pre-target fixations was not due to a contamination from target fixations.

**No confounding factor from the orthographic information.** One potential concern for the current study is the influence of orthographic information. First, we found that bi/trigram type frequency did not co-vary with word frequency, but bi/trigram token frequency and neighborhood size did (see Supplementary Table 1). Next, we qualified a possible orthographic parafoveal processing effect by separating trials based on these three co-varying orthographic variables. All parameters and procedures were the same as in the lexical parafoveal analysis (Fig. 4). However, no significance was found for bi/trigram token frequency or neighborhood size (Supplementary Fig. 6; $p$ values were 0.69, 0.06, and 0.44 seperately, Bonferroni corrected). Please note that trigram token frequency co-varied greatly with word frequency, which could explain the marginal significance. We

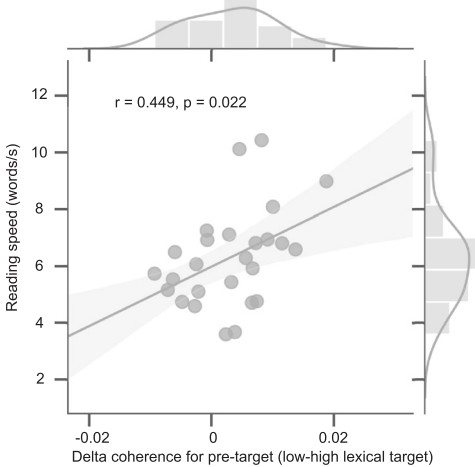

**Fig. 5 Relation between lexical parafoveal processing difference and individual reading speed.** The coherence-difference with respect to target lexical frequency during fixations on pre-target words were derived per participant (see Fig. 4b). The reading speed was quantified as the number of words read per second. Each dot represents a participant. A Spearman correlation demonstrated a significant relation ($n = 26$, two-sided). The solid gray line represents the linear regression fit with a 95% confidence interval (shaded area). Histograms of individual reading speed and pre-target coherence differences are shown on the top and to the right. Source data are provided as a Source Data file.

conclude that orthographic information is not a significant confounding factor for the reported neural effects on lexical parafoveal processing effects.

**Lexical parafoveal processing facilitates reading.** We correlated the pre-target coherence difference (low minus high lexical target frequency) with individual reading speed. Reading speed was quantified as the number of words read per second (i.e., the number of words in all sentences divided by the total reading time). We found a positive correlation (Fig. 5; $r_{(25)} = 0.45$, $p = 0.022$, Spearman correlation), indicating that participants who captured more lexical information in the parafovea were also faster readers.

**Late lexical parafoveal effect observed from fixation-related fields.** In order to investigate if the lexical parafoveal effect could also be observed in fixation-related fields (FRFs), we compared FRFs for pre-target words followed by low and high lexical frequency target words over all participants ($n = 39$).

We conducted a cluster-based permutation test over all combined planar gradiometers (0–0.5 s, aligning with fixation onset for pre-target words). We found clusters of sensors that had significantly higher FRFs when the pre-target word was followed by low compared with high lexical target words (Fig. 6a, Pcluster < 0.05, two-tailed pairwise $t$-test, 1000 permutations). Averaged pre-target FRFs over these significant sensors are shown in Fig. 6b. We observed the strongest effect around 0.4 s in the left posterior sensors. We formed the same FRFs analysis for target fixations but we did not find a significant difference (see Supplementary Fig. 7 for the target averaged FRFs over sensors shown in Fig. 6a).

**Discussion**
Our findings shed light on the long-standing debate between models arguing for either serial attention shift[8,9] or parallel graded processing[16,17]. Both models regard spatial attention as

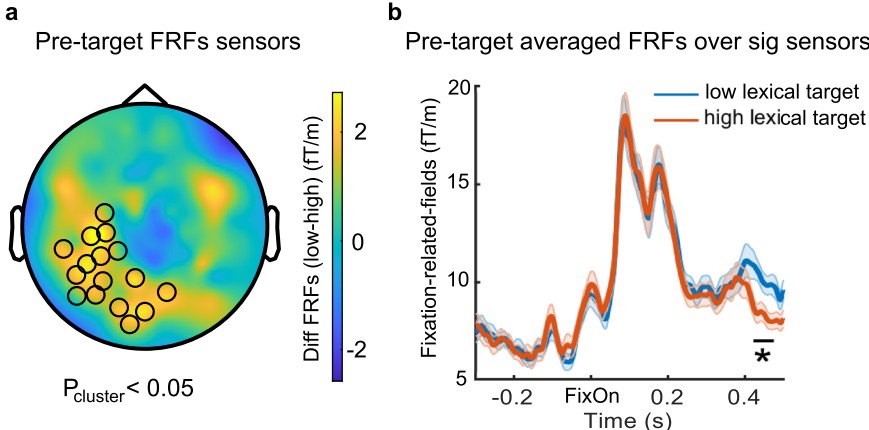

**Fig. 6 Fixation-related fields for pre-target words. a** Topography for the sensors that showed a lexical parafoveal processing effect during pre-target fixations. The color bar indicates the FRFs difference for pre-target words followed by low compared with high lexical target words ($n = 39$, cluster-based permutation with $p < 0.05$, two-tailed pairwise $t$-test). **b** Pre-target averaged FRFs over these significant sensors for low (solid blue line) and high (solid orange line) lexical target conditions. The shaded areas represent standard error over 39 participants.

important for reading but they differ on whether more than one word can be accessed simultaneously at the lexical level. Parallel graded models predict that both foveal and parafoveal words can be processed lexically simultaneously. This idea has been challenged by the fact that only a few correlation-based corpus studies found lexical parafoveal effects in eye movement data[31,32], while more well-controlled experimental studies did not[19–21] (for a meta-analysis study see ref. [33]), and neither did our eye movement results (Fig. 2). These eye movement data have been used to argue in favor of serial attention shift models. In this study, we found neural evidence for lexical parafoveal effects as early as 100 ms from pre-target onset (Fig. 4b). This finding does not seem to be compatible even with the most temporally compressed version of a serial model, in which the attentional shift and a significant amount of lexical parafoveal processing occur during saccadic programming[34–36]. Moreover, evoked response studies based on EEG[37,38] and MEG[39] provide evidence for lexical frequency effects for fixated words no earlier than 100 ms. In conclusion, our results showed a lexical frequency effect for the parafoveal word at around 100 ms, which supports the idea of parallel processing of foveal and parafoveal information.

One might ask why lexical parafoveal processing is reflected in neuronal responses (Fig. 4b) but not in fixation durations (Fig. 2). We would like to stress that albeit lexical parafoveal processing was not reflected in the pre-target fixation durations, the neuronal effects were linked to reading speed. Basically participants who read faster also have a stronger parafoveal lexical neuronal modulation. This suggests that parafoveal processing is reflected by the allocation of covert attention, especially to less common targets words. However, this allocation of covert attention does not directly impact overt attention, i.e., the decision criteria for when to initiate the saccade. Possibly, the absence of lexical parafoveal effects on pre-target fixation times might facilitate fluent reading. Prolonging the current fixation when previewing a difficult word is not an efficient strategy, since it means keeping the difficult word in the low visual acuity parafovea for longer. Taken together, our study shows that natural reading involves the simultaneous processing of several words in a graded way, providing neural evidence for the idea that "readers are parallel processors"[40].

Why did we not find differences in neuronal excitability with respect to lexical effects for the foveal target words, even though previous studies have shown that RIFT can be used to investigate both foveal[27,28] and parafoveal processing[25,26]? In these previous studies, the flickering stimuli are larger (at least 5.7° visual angle in diameter) compared to the size of the flickering words used in this study (at the most 3° by 1° visual angle). Also the durations to perceive the flicker were longer (at least 1.5 s) compared with here (around 0.2 s). In addition, flicker sensitive photoreceptors (rod cells) are more abundant in the parafovea of the retina[41]. As a consequence, there would be higher sensitivity to flickering in the parafovea compared to the fovea.

The neuronal response reflecting lexical parafoveal processing was observed in the early visual cortex. This might be a surprise, as functional magnetic resonance imaging studies have localized lexical frequency to e.g., the visual word form area[42]. According to interactive processing theories, higher-level lexical information interacts with lower-level visual information during word recognition[37,43], and the feedback modulation can be measured by MEG over sensory cortices[44]. Thus, lexical frequency information extracted in the parafovea could direct visual attention covertly. Increased spatial attention will boost tagging responses[25,26], resulting in stronger coherence for the pre-target word followed by a low compared with a high lexical frequency target word. We interpret the stronger frequency tagging for low-frequency target words as being a consequence of the allocation of more covert attention to the parafoveal word. The covert attention will help to facilitate the processing of less familiar words. Alternatively, one might have expected high-frequency words to be more attention grabbing as they are more familiar; however, the frequency tagging result suggest that this is not the case.

Our results show that RIFT is a powerful technique to investigate parafoveal reading. A classic paradigm in this field is the gaze-contingent boundary task developed by Keith Rayner in 1975[45]. In this task, parafoveal information is manipulated by changing the target word while saccading to it[20,46]. This approach allows for manipulating parafoveal processing and has made great contributions to studies on parafoveal processing (for reviews see refs. [7,47]). However, the approach is limited, as changing the target word inevitably disrupts the integration of information across fixations and interferes with natural reading. This interference has been shown in many gaze-contingent studies in which reading performance is reduced when words are manipulated in the parafovea[20,48–50]. Fixation or event-related potentials based on EEG is another method used in reading studies[51–54], but have shown different brain activity patterns for different word presentation rates[55], addressing the importance of using natural reading paradigms. The importance of natural reading paradigms

is also supported by an MEG study that found different neural patterns when the priming word was in the fovea and parafovea, where the latter is relevant to natural reading[56]. While fixation-related potentials have provided important insights by demonstrating a lexical frequency effect for foveal word recognition on the N1 component[37,38], it has failed to provide conclusive results with regard to parafoveal lexical parafoveal. Some studies did not find evidence for lexical parafoveal in the FRPs[20,21], while one other study[29] found this effect around 400 ms, compatible with our findings (see Fig. 6b). This late effect probably reflects the integration of target words into prior context. In comparison, the RIFT approach allowed us to capture parafoveal processing at a much earlier stage of word processing. As such the FRFs/FRPs and the RIFT approach provide complementary information about word processing during natural reading.

It would be interesting to use the RIFT approach to investigate whether parafoveal processing might also occur at higher levels, such as extracting semantic information as suggested by natural reading studies of Chinese[57] and German[58]. Similarly, parafoveal processing at the syntactic level[59] would also be of great interest to investigate. Another direction is investigating the primary determinants of reading proficiency in relation to parafoveal processing. For instance, parafoveal processing at the phonological and orthographic level[60] has been found to reflect reading proficiency. We found that the neuronal signature of lexical parafoveal predicted reading speed, which could be used as a potential indicator to diagnose reading disorders such as dyslexia. Some researchers argue that dyslexia is due to spatial processing problems in the magnocellular visual pathway[61], as shown in an MEG study[62]. Our frequency tagging approach could be helpful to understand the underlying neural mechanism of dyslexia in relation to the allocation of spatial attention.

In sum, the present study demonstrates that RIFT is a powerful tool for investigating natural reading, and provides neural evidence for lexical parafoveal processing in support of parallel graded models of reading.

## Methods

**Participants**. Our study recruited forty-three participants (28 females), aged $22 \pm 2.6$ (mean ± SD), right-handed, with normal or corrected-to-normal vision, and without a neurological history or language disorder diagnosis. Four of them were excluded from analysis due to poor eye tracking or falling asleep during the recordings, which left thirty-nine participants (25 females). The University of Birmingham Ethics Committee approved the study. The participants provided written informed consent and received £15 per hour or course credits as compensation for their participation.

**Stimuli**. For the full list of all the sentences that were used in this study, please see Supplementary Methods.

*1st sentence set*. We constructed 142 sentences embedded with 71 target word pairs (low/high lexical frequency). For each sentence, the pre-target, target, and post-target words were in the same structure as adjective + noun + verb. For each target pair, two different sentence frames were made, and each participant read both target words embedded in these two different frames. For example, for the target pair **waltz/music** (low/high lexical frequency), one participant read version A, another one read version B (see below, targets are in bold for illustration, but not in the real experiment).

- A. Mike thought this difficult **waltz** received lots of criticism.
  It was obvious that the beautiful **music** captured her attention.
- B. Mike thought this difficult **music** received lots of criticism.

It was obvious that the beautiful **waltz** captured her attention.
The sentences in version B were made from version A by circular shifting the first and second half of the sentences. For both versions, no more than three successive sentences were from the same target lexical frequency condition.

*2nd sentence set*. This sentence set was adapted from Degno et al.[20]. We removed sentences that contained the same pre-target or target words as in the 1st sentence set, which left 86 sentences. Each sentence was embedded with two target words from the same lexical frequency condition (see below, version A contained two low

lexical frequency targets, while version B contained two high lexical frequency targets).

- A. I felt quite **bleak** after discussing that really **risky** subject with Paul.
- B. I felt quite **weird** after discussing that really **nasty** subject with Paul.

Each participant read either version A or B. The same control for sentence presentation was counterbalanced as in the 1st set.
We conducted pre-tests with another group of participants to make sure all sentences were plausible with either low-lexical or high-lexical frequency target word, and that the target words were not predictable (see Supplementary Methods).

**Procedure**. Participants were seated comfortably in the MEG gantry, 145 cm away from the projection screen in a dimly-lit magnetically shielded room. One-line sentences were presented on a middle-gray screen using Psychophysics Toolbox-3[63]. Every sentence started at the same position: two degrees to the right of the middle of the screen left edge and was presented on the vertical center. Words were displayed in black font color with an equal-spaced Courier New font (size 22). Each letter and space between two words occupied 0.35 visual degrees. In total, no sentence was longer than 27 visual degrees horizontally. Two sets of sentences, consisting of respectively 142 and 86 sentences, were divided into seven blocks. Each block took approximately 7 min to read and was followed by a rest for at least 1 min. Participants were instructed to read each sentence silently at their own pace and to keep their head and body as stable as possible during the MEG session. Eye movements were acquired during the whole session.

Each trial started with a central fixation cross on a gray screen center presented for 1.2–1.6 s. Then followed by a square (1° wide) presented 2° to the right of the middle of the screen left edge. A gaze of at least 0.2 s on this square triggered sentence onset. The square was replaced by the first word of the sentence. The text was presented in the equal spaced Courier New font, and each letter occupied 0.35 visual degrees (Fig. 1). After reading the sentence, participants were instructed to fixate on a square below the screen center for 0.1 s to trigger the sentence offset. One-quarter of the trials were followed by a simple yes-or-no comprehension question to ensure careful reading. All participants answered the questions with high accuracy (95.4 ± 4.7%, mean ± SD).

## Rapid invisible frequency tagging

*Projector*. To generate the rapid invisible frequency tagging, sentence stimuli were presented with a refresh rate up to 1440 Hz using a PROPixx DLP LED projector (VPixx Technologies Inc., Canada). This was done by presenting the sentence stimuli repeatedly in four quadrants on the stimulus computer screen (1920 × 1200 pixels resolution) with a refresh rate of 120 Hz. For each quadrant, the stimuli were coded in RGB three color channels. The projector interpreted these 12 color channels (3 channels × 4 quadrants) as 12 individual grayscale frames and projected them onto the projector screen separately in rapid succession. Hence, the refresh rate for stimuli presentation was 1440 Hz (120 Hz × 12).

*Flickering of the target word*. To flicker the target word, we added a rectangular patch underneath the target. The width of the patch was the width of the target word plus the spaces on both sides (2–3° visual angle). The height of the patch was 1.5 times the word height (1° visual angle). The target word was placed in the center of this rectangular patch. All pixels within the patch were flickered at 60 Hz by multiplying the luminance of the pixels with a 60 Hz sinusoid (the modulation depth was 100%). Typically, the patch was perceived as indistinguishable from the middle-gray screen background, which made it invisible to participants. To reduce the visibility of the patch edges during saccades, a Gaussian smoothed transparent mask was applied on top of the flickering patch. The mask was created by a two-dimensional Gaussian function (Eq. 1):

$$\text{Mask} = \exp\left(-\frac{x^2}{2\sigma^2} - \frac{y^2}{2\sigma^2}\right), \tag{1}$$

where $x$ and $y$ are the mesh grid coordinates for the flickering patch, and $\sigma$ is the $x$ and $y$ spread of the blob with $\sigma = 0.02°$. By applying a Gaussian smoothed mask, the flickering patch was hardly perceived. Only three out of all the thirty-nine participants noticed the flickering patch according to a questionnaire after the MEG session.

A custom made photodiode (Aalto NeuroImaging Center, Finland) was attached to the right-below corner of the screen to record tagging signal from a square whose luminance was kept the same as the flickering patch.

## Data acquisition

*MEG*. MEG data were acquired using a 306-sensor TRIUX Elekta Neuromag system with 204 orthogonal planar gradiometers and 102 magnetometers (Elekta, Finland). The data were band-pass filtered online using anti-aliasing filters from 0.1 to 330 Hz and then sampled at 1000 Hz. We used a Polhemus Fastrack electro-magnetic digitizer system (Polhemus Inc., USA) to digitize the locations for three bony fiducial points: the nasion, left and right preauricular points. Then, four head-position indicator coils (HPI coils) were digitized: two coils were attached on the left and right mastoid bone and another two were on the forehead with at least 3 cm distance in between. Furthermore, at least 200 extra points on the scalp were

acquired for each participant in order to spatially co-register the MEG source analysis with individual structural MRI image. After preparations, participants were seated upright under the MEG gantry with the back rest at a 60°angle.

*Eye movements.* The eye-tracker (EyeLink 1000 Plus, SR Research Ltd, Canada) was placed on a wooden table in front of the bottom edge of the projector screen. The distance between the eye-tracker camera and the center of the participant's eyes was 90 cm. It was used throughout the whole experiment to acquire horizontal and vertical positions of the left eye as well as the pupil size. Eye movements were sampled at 1000 Hz. We also placed one pair of electrodes above and below the right eye (vertical electrooculogram, EOG) and another pair to the left and right of the eyes (horizontal EOG) to provide additional measures for ocular and eye-blink artefacts.

Each session began with a nine-point calibration and validation test. After every three trials, we performed a one-point drift checking test. If a participant failed to pass drift checking or was unable to trigger sentence onset through gazing, the nine-point calibration and validation test was conducted again. The eye-tracking error was limited to below 1 visual degree both horizontally and vertically.

*MRI.* After MEG data acquisition, we acquired the T1-weighted structural MRI image using a 3-Tesla Siemens PRISMA scanner (TR = 2000 ms, TE = 2.01 ms, TI = 880 ms, flip angle = 8°, FOV = 256 × 256 × 208 mm, 1 mm isotropic voxel). Out of all the thirty-nine participants, three dropped out of the MRI acquisition, one of them showed robust tagging responses at the sensor level. For this participant, the MNI template brain (Montreal, Quebec, Canada) was used instead in later source analysis.

**MEG data analyses.** The data analyses were performed in MATLAB R2019b (Mathworks Inc., USA) by using the FieldTrip[64] toolbox (version 20200220) and custom-made scripts.

*Pre-processing.* The MEG data were band-pass filtered from 0.5 to 100 Hz using phase preserving two-pass Butterworth filters. First, the MEG segments were extracted from −0.5 to 0.5 s intervals aligned with the first fixation onset for pre-target, target, and post-target words, respectively. Only segments with fixation durations ranging from 0.08 to 1 s entered further analyses. Segments with too short or too long fixations were discarded. We also extracted 1 s long baseline segments aligned with the presentation onset for the cross-fixation, which was the period before sentence onset. Next, the MEG data were demeaned by removing the linear trend and the mean value. After removing malfunctioning sensors (0–2 sensors per participant), these segments entered an independent component analysis (ICA)[65]. Data were decomposed into independent components with the same number of sensors. Next, the components related to eye blinks, eye movements, and heartbeat were rejected. Finally, we manually inspected all these segments to further identify and remove any segments that were contaminated by excessive noise like ocular, muscle, or movement artefacts.

*Coherence calculation.* To measure the tagging response associated with target word processing, coherence was estimated between the MEG sensors and the tagging response of the photodiode (for MEG sensor selection see below). First, 1 s segments were filtered using a phase preserving, two-pass, Butterworth bandpass filters (4th order) with a hamming taper. The center filter frequencies were from 40 to 80 Hz in steps of 2 Hz with a 10 Hz frequency smoothing. For each frequency step, the analytic signals were determined by the Hilbert transform which then was used as the input for coherence at time point $t$ (Eq. 2):

$$\text{coh}(t) = \frac{\left| n^{-1} \sum_{j=1}^{n} m_{x_j}(t) m_{y_j}(t) e^{i \varnothing_{xy_j}(t)} \right|^2}{n^{-1} \sum_{j=1}^{n} m_{x_j}(t)^2 m_{y_j}(t)^2}, \qquad (2)$$

where $j$ is the trial, $n$ is the number of trials, $m_x(t)$ and $m_y(t)$ are the time-varying magnitude of the analytic signals from respectively a MEG sensor and a photodiode, $\varnothing_{xy}(t)$ is the phase difference as a function of time between them. A time-frequency coherence representation was obtained as applied in Figs. 3c, 4a, c.

*RIFT response sensor selection.* To identify the MEG sensors that showed reliable tagging responses, we compared the 60 Hz coherence during pre-target segments with the coherence during baseline segments. We used a non-parametric statistics method named Monte-Carlo to estimate the significance for the coherence difference. This method was developed by Maris et al.[66], and implemented in the Fieldtrip[64] toolbox. Both pre-target and baseline segments were 1 s long and were aligned with the first fixation onset for pre-target words and the onset for baseline cross-fixation separately. The pre-target segments were constructed by pooling the target lexical frequency conditions together. Several previous RIFT studies from our lab observed robust tagging responses from the visual cortex for visual flickering stimuli[25–28]. Therefore, only MEG sensors in the visual cortex (52 planar sensors) entered this sensor selection procedure.

Here, we regarded pre-target and baseline segments as two conditions in the coherence calculation. For a given MEG sensor and photodiode combination, coherence at 60 Hz was estimated over trials for each condition. Therefore, one coherence value was obtained for each condition. Then, we calculated the z-statistic value for this coherence difference between pre-target and baseline using the following equation (for details please see Maris et al.[66]) (Eq. 3):

$$Z = \frac{(\tanh^{-1}(|\text{coh}_1|) - \text{bias}_1) - (\tanh^{-1}(|\text{coh}_2|) - \text{bias}_2)}{\sqrt{\text{bias}_1 + \text{bias}_2}},$$
$$\text{bias}_1 = \frac{1}{2n_1 - 2}, \text{bias}_2 = \frac{1}{2n_2 - 2}, \qquad (3)$$

where $\text{coh}_1$ and $\text{coh}_2$ denote the coherence value for pre-target and baseline condition, $\text{bias}_1$ and $\text{bias}_2$ is the term used to correct for the bias from trial numbers of pre-target ($n_1$) and baseline condition ($n_2$). So, all trials from the pre-target and baseline conditions were used. After obtaining the z statistic value for the empirical coherence difference, a permutation procedure was conducted to estimate the significance probability.

After obtaining the z-scored values for the empirical coherence differences, a permutation procedure was conducted to estimate the statistical significance. We randomly shuffled the trial labels between pre-target and baseline conditions 10,000 times. During each permutation, coherence was computed for both conditions (with shuffled labels), then entered in the above formula to obtain a z-score value for the randomization procedure. After all the shuffles, a null distribution for z-values was established. If the empirical z-value was larger than 99% of the null distribution, which meant that the coherence difference between pre-target and baseline was larger than zero at the 0.01 significance level, this sensor was considered to have robust tagging responses. This sensor selection procedure was performed for every sensor in the visual cortex (52 planar sensors in total). Twenty-six out of all the thirty-nine participants showed robust tagging responses at one or more sensors (5.4 ± 4.0 sensors per participant, mean ± SD, Fig. 3a; for tagging sensors for each participant please see Supplementary Fig. 2). For each participant, the coherence values were averaged over all tagging response sensors to obtain an averaged coherence.

*Source analysis for RIFT.* In order to localize the neural sources that were coherent with the photodiode signals during RIFT, a beamforming approach was performed using Dynamic Imaging Coherent Sources (DICS)[67] implemented in the FieldTrip[64] toolbox. The DICS technique enabled us to calculate the source estimates in the frequency domain with a focus on 60 Hz, which was the RIFT frequency. The beamformer was based on adaptive spatial filters derived for each grid in the discretized brain volume. In this source analysis, only participants with robust tagging responses were included ($n = 26$; see Fig. 3a).

A semi-realistic head models was constructed using a procedure developed by Nolte[68], which uses spherical harmonic functions to fit the brain surface. We first aligned the individual structural MRI image with the MEG data. This was done by spatially co-registering the three fiducial anatomical markers from the head shape digitization during the MEG session (nasion, left, and right ear canal). For one participant whose MRI image was unavailable, the MNI template brain was used instead. Next, this aligned MRI image was segmented into a grid. Then, we prepared the single-shell head model based on the segmented MRI image.

The individual source model was constructed by inverse-warping a 5 mm spaced regular grid in the MNI template space to each participant's segmented MRI image in the native space. This regular grid was from the Fieldtrip template folder and was constructed before doing the source analysis. In this way, the beamformer spatial filter was constructed on the direct grid that mapped to the MNI template space.

The cross-spectral density (CSD) matrix was calculated at 60 Hz between all possible combinations chosen from the MEG sensors and photodiode (MEG sensor + MEG sensor as well as r MEG sensor + photodiode). This was done for each participant during the 1 s long pre-target and baseline segments, after using a hamming taper. No regularization was performed to the CSD matrices (lambda = 0).

Next, a common spatial filter was computed based on the individual single-shell head model, source model, and CSD matrices using DICS. This spatial filter was applied to both the pre-target and baseline CSD matrices for coherence computation. This was done by normalizing the magnitude of the summed CSD between the MEG sensor and the photodiode by their respective power. After the grand average over participants, the relative change for pre-target coherence was estimated as the ratio between coherence difference and baseline coherence (($\text{coh}_{\text{pretarget}} - \text{coh}_{\text{baseline}})/\text{coh}_{\text{baseline}}$). Finally, this source analysis localized the RIFT neural sources to the left-visual associate, Brodmann area 18, MNI coordinates [−4 −97 3] (see Fig. 3b, $n = 26$).

*Fixation-related fields analysis.* This analysis was performed for all participants ($n = 39$). For the 1 s pre-target epochs (−0.5 to 0.5 s with fixation onset to the pre-target word), we applied a 35 Hz low-pass filter using phase preserving two-pass Butterworth filters. For each participant, the same number of trials were randomly selected based on the minimum trial number across the low-frequency and high-frequency conditions. After grand averaging the epochs with a baseline subtraction

(−0.2 to 0 s), we obtained fixation-related fields (FRFs) for pre-targets in both conditions. For each pair of planar sensors at the same location, we combined the root-mean-square of the FRPs.

A permutation test was conducted for the averaged pre-target FRFs over all valid sensors to compare the lexical effects. A cluster-based permutation test was conducted for the whole time window (0 to 0.5 s) with a threshold of $p = 0.05$ (two-tailed pair-wise $t$-test). To form a cluster required at least two neighboring sensors that exceeded the threshold. The neighborhood layout for sensors was pre-defined by FieldTrip for the MEGIN system. Next, we created a reference distribution by permuting the data 1000 times. In each permutation, the labels for low and high target conditions were randomly assigned over participants, and the $t$-values are calculated to obtain the clusters. The maximum summed $t$-value from each cluster was used to construct the reference distribution. After 1000 permutations, we sorted all $t$-values in the null distribution from the minimum to the maximum. $T$-values at the 25th and the 975th position were chosen as the thresholds for negative (FRFs_high > FRFs_low) and positive (FRFs_low > FRFs_high) clusters in the experimental data. We only found significant positive clusters ($p < 0.05$), indicating that FRFs were higher for pre-target words followed by low compared with high lexical target words.

**Statistical information**. All the $t$-tests in this study were two-sided pairwise student's $t$-tests and were conducted in R[69].

**Reporting summary**. Further information on experimental design is available in the Nature Research Reporting Summary linked to this paper.

## Data availability
Source data are provided with this paper. We have deposited the following data in the current study on figshare (https://figshare.com/projects/Pan_etal_NatCom_2021/117885)[70]: the raw MEG data, the epoch data after pre-processing, the raw EyeLink files, the Psychtoolbox data, and the head models after the co-registration of T1 images with the MEG data. The raw T1 images are not shared due to sensitive personal information (faces). De-identifying T1 images will remove the informative facial landmarks and make it difficult to construct head models. Therefore, we share the head models instead of the T1 images. Any additional information will be available from the authors upon reasonable request. Source data are provided with this paper.

## Code availability
The experiment presentation scripts (Psychtoolbox), statistics scripts (R), scripts and data to generate all figures (Matlab) are available on OSF (https://doi.org/10.17605/OSF.IO/ARD6H)[71].

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

## Acknowledgements
We thank Dr. Federica Degno and Prof. Simon Liversedge for sharing the second sentence set, Dr. Geoffrey Brookshire and Yang Cao for feedback on the manuscript. This work was supported by the following grants to O.J.: the James S. McDonnell Foundation Understanding Human Cognition Collaborative Award (grant number 220020448), Wellcome Trust Investigator Award in Science (grant number 207550), and the BBSRC grant (BB/R018723/1) as well as the Royal Society Wolfson Research Merit Award.

## Author contributions
Y.P., S.F. and O.J. devised and designed the experiments, Y.P. made the sentences with assistance from S.F., Y.P. programmed and conducted the experiments, Y.P. carried out the analyses with assistance from O.J. and S.F., Y.P., O.J. and S.F. wrote the paper together.

## Competing interests
The authors declare no competing interests.
