## [Peer Review File · Nature Communications]

Neural evidence for lexical parafoveal processingREVIEWER COMMENTS

Reviewer #1 (Remarks to the Author):

Review on Pan et al.: Lexical parafoveal previewing predicts reading speed

The manuscript reports neural evidence in support of ongoing lexical parafoveal previewing during reading—using an innovative experimental approach combining RIFT, MEG, and eye-tracking. The writing is clear and succinct, analyses are sound, and conclusions are generally reasonable. I congratulate the authors for this excellent work which adds important new knowledge to our understanding of reading, one of the most critical human skills in modern society.

Comments:

1. (line 39) Parallel graded attention models. Besides (14, 15) there is another relatively recent model (OB1 reader, Snell et al., 2018), which the author might consider to cite in addition.
2. (line 43) The statement that impact of the lexical frequency of the upcoming parafoveal word (parafoveal-on-fovea/POF effect) has not been observed is generally correct. However, eye movement recording still represents a powerful tool for evaluation of models. For example, while Ref. (20) does not present evidence for a POF effect, the authors argue that there is a delayed fixation duration effect observable after the boundary. Thus, there are effects that cannot be squared with the strict serial attention model. The bottom line from this comment is that eye movements lend support for parafoveal lexical processing if interpreted with care.
3. (line 99) Mislocated fixations have been put forward as an explanation for apparent POF effects in the strict serial attention model, however, this cannot explain the effect because of the so-called inverted-optimal viewing position effect of fixation duration (Vitu et al., 2001; Nuthmann et al., 2005).
4. (line 206) It should also be mentioned that there are syntactic effects from the parafovea (Snell et al., 2017), which might be even more higher-level language processing based on parafoveal information. Also, it should be mentioned that there is evidence for semantic parafoveal-on-fovea effects in Chinese reading (Yan et al., 2009).

Refs.

- Snell, J., van Leipsig, S., Grainger, J., & Meeter, M. (2018). OB1-reader: A model of word recognition and eye movements in text reading. *Psychological Review*, 125(6), 969-984.
<http://dx.doi.org/10.1037/rev0000119>
- Yan, M., Richter, E. M., Shu, H., & Kliegl, R. (2009). Readers of Chinese extract semantic information from parafoveal words. *Psychonomic Bulletin & Review*, 16(3), 561-566.
<https://doi.org/10.3758/PBR.16.3.561>
- Vitu, F., McConkie, G. W., Kerr, P., & O'Regan, J. K. (2001). Fixation location effects on fixation durations during reading: An inverted optimal viewing position effect. *Vision Research*, 41(25-26), 3513-3533. [https://doi.org/10.1016/S0042-6989\(01\)00166-3](https://doi.org/10.1016/S0042-6989(01)00166-3)
- Nuthmann, A., Engbert, R., & Kliegl, R. (2005). Mislocated fixations during reading and the inverted optimal viewing position effect. *Vision Research*, 45(17), 2201-2217.
<https://doi.org/10.1016/j.visres.2005.02.014>
- Snell, J., Meeter, M., & Grainger, J. (2017). Evidence for simultaneous syntactic processing of multiple words during reading. *PloS one*, 12(3), e0173720.
<https://doi.org/10.1371/journal.pone.0173720>

EOF

Reviewer #2 (Remarks to the Author):

This MEG study investigated parafoveal preview effects in sentence reading using a novel Rapid Invisible Frequency Tagging (RIFT) paradigm. The target word in a visually presented sentence was flickered at a frequency of 60Hz. The eye movements during natural sentence reading were monitored using eye-tracking. The coherence between the flicker signal and the brain response during the fixation of the pre-target word was the main dependent variable. Its modulation by the lexical frequency of the target word is interpreted as evidence that readers access lexical information of the upcoming word parafoveally.

This study addresses a long-standing question in psycholinguistic reading research using a novel and innovative experimental paradigm. Previous research has shown that lexical parafoveal preview effects on behaviour and brain responses, if they exist at all, are subtle at best. The current study adds new empirical data to this debate. The authors are correct in saying that previous research has failed to provide conclusive results with regard to parafoveal lexical processing. We think that the authors' approach provides an interesting new angle on this issue, but as the effects are subtle and several aspects of this study are unclear to us, we are not yet convinced that this new evidence is conclusive either.

1) The authors discuss the fact that they found a subtle preview effect on coherence values but not on behaviour (ll. 219ff., "One might ask why..."). This raises the question whether the observed effects, if reliable, really reflect the "simultaneous processing of several words" in a way that is relevant for the cited computational models, or a separate "detection process", which may not be relevant for reading behaviour (and therefore not within the scope of those models, but possibly interesting for other reasons). The authors attempt an explanation in terms of "attracting more or less attention", but if this is a valid explanation (which we think was also the prediction in previous behavioural studies on this topic) then one would indeed expect behavioural effects? Maybe the RIFT is more sensitive than behaviour. Is this something that could be analysed in more detail, e.g. in terms of power analysis and comparing number of stimuli and participants with previous studies?

2) Coherence for the pre-target fixation was stronger for low compared to high frequency target words. This is not discussed, even though it may support the "attention explanation" of the lexical preview effects: low frequency words are harder to process and require more attention. On the other hand, high frequency words may grab more attention (in this case for example in order to trigger an earlier saccade). This would fit better with the attention literature, where more salient and more familiar stimuli are usually easier to detect and more distracting. This would deserve a more detailed discussion. And again, these explanations would also lead to predictions for behavioural effects.

3) The authors provide an explanation for the absence of tagging effects for target word fixations (l. 179). Does this mean this paradigm can only be used to study parafoveal vision? If we understood correctly, at least the cited Drijvers et al. HBM 2020 study used frequency tagging for foveal stimuli.

4) These results would be much stronger if the authors could replicate previously reported word frequency effects for target word fixations on EEG/MEG data in a conventional evoked analysis.

5) The latency analysis of the preview effect is interesting (ll. 154ff.). However, an effect is first "observed" but then only found "robust" in one dataset, which is attributed to word length effects. The word length hypothesis could be tested as an interaction with word length. The presence of an effect in only one dataset, without further justification or an interaction, is not strong evidence for anything. This analysis could be described as exploratory, and possibly moved to the supplementary section.

6) A more complete set of psycholinguistic variables for the target and pre-target words must be provided. Table 1 does not show 'length' and 'position' for low and high frequency words separately, and it does not show important orthographic variables such as bi/trigram frequency and Coltheart's neighbourhood size or orthographic Levenshtein distance, which are important orthographic confounds for possible preview effects.

7) Source estimation results are presented for the tagging response, but not for the frequency effect. The authors then interpret the results as if they are specific to visual cortex. This interpretation would be stronger with a separate whole-brain or ROI analysis.

8) The main analysis was run on 26 out of 39 participants who showed a significant tagging response (ll. 121ff.). This is fine, but shows that this response is rather subtle. Is this common practice in this kind of experiment? We could not find this kind of information in Drijvers et al. HBM 2020 and Zhigalov et al. NI 2019. Is the absence of tagging responses in one third of the participants assumed to be mostly physiological/anatomical in nature, or could this point to individual differences in reading behaviour? For example, could the non-responders still be included in the correlation analysis?

9) Methodological clarifications:

a. PCA was used to reduce the rank of the data to 30 components (l. 408). Is this a common approach in this field (it also appears to be different from the other two frequency tagging studies)? If the subtle experimental effects are reflected in the lower components then this approach may lose information.

b. Please provide some more details on how the covariance/cross-spectral matrices for the beamforming analysis were computed and regularized. Was the reduction in rank taken into account in the construction of the beamformer?

c. The authors use “coherence” and “r²” in different places. This should be consistent, e.g. “coherence” throughout. We understand the rationale for using coherence. However, can the phase of the tagging response be assumed to be stable at this (relatively high) frequency? Would it be interesting to run a separate analysis on signal power at the tagging frequency (as in Zhigalov et al. 2019)?

d. It wasn't clear to us which exact latency ranges were used for the statistical frequency tagging analysis (ll. 144ff.).

Reviewer #3 (Remarks to the Author):

Review of manuscript entitled “Lexical parafoveal previewing predicts reading speed” (NCOMMS-21-04998) by Pan, Frisson, and Jensen.

Summary. The manuscript describes an eye-tracking experiment in which high- vs. low-frequency targets words in the parafoveal were “flickered” at 60 Hz (using RIFT) to determine if the neuronal activity induced by this manipulation could be measured (using MEG) from the pre-target word. This manipulation did produce a frequency-modulated neuronal response for short words (from one of two sets of sentences), but interestingly, target-word frequency did not modulate fixations on the pre-target words (i.e., there was no evidence for parafoveal-on-foveal effects). The former effect was also correlated with reading speed; as reading speed increased, the size of the MEG-measured neuronal activity increased. On the bases of these findings, the authors argue that their study is more consistent with attention-gradient models of reading, wherein multiple words are subject to concurrent lexical processing, rather than serial-attention models, wherein only one word is subject to lexical processing at any given time.

General comments. I really enjoyed reading this manuscript. The manuscript itself was well written, and the experiment was innovative and appeared to be methodologically sound. (Please note, though, that I'm not an expert on RIFT or MEG). Possibly offsetting these strengths, however, is the fact that the experimental results are open to an alternative interpretation—one that is actually consistent with the serial-attention models that the authors argue against. My concerns about the manuscript (both major and minor) are outlined below.

Major Concerns/Questions:

1. p. 2, lines 34-35: The authors claim that, according to serial-attention models, “lexical information is

not generally extracted from the parafovea.” This statement is incorrect. For example, according to the E-Z Reader model, most word skipping reflects instances in which the parafoveal word has received sufficient lexical processing to cancel the saccade that would otherwise cause the word to be fixated (Reichle & Drieghe, 2015). Similarly, simulations using E-Z Reader (Schotter, Reichle, & Rayner, 2014) indicate that, within the framework of a strictly serial-attention model, significant time is available for late lexical (e.g., semantic) processing of upcoming words.

2. p. 5. The authors report a fixation landing-position distribution (Fig. 2b) to argue against the notion that their observed effect of parafoveal flicker was due to mis-located fixations (i.e., pre-target fixations intended for the targets but that fell short due to saccadic error). Unfortunately, this analysis fails to rule out the possibility of mis-located fixations: Because each letter subtended 0.35 visual degrees (p. 15) but calibration error is up to 1 visual degree (p. 18), I think it’s reasonable to assume that some proportion of the pre-target fixations were intended for the targets, suggesting that overt attention (as measured by the flicker manipulation) may have already been focused on the targets. Simulations have demonstrated how purported parafoveal-on-foveal effects can arise due to saccadic error and/or eye-tracking measurement error even in a strictly serial-attention model (Reichle & Drieghe, 2015).

3. p. 6, line 129: The source localization analyses suggest that the flicker effect was “localized in the early visual cortex.” Later in the manuscript (p. 12), the authors acknowledge that this is at odds with what might have been predicted (e.g., word-frequency effects are generally localized in the visual word-form area), but then argue that interactive models might account for this finding via “top-down modulation ... over sensory cortices.” This account seems ad hoc and lacks parsimony. A more parsimonious account might simply be that the experiment was successful at demonstrating concurrent visual processing of foveal and parafoveal words—a finding that aligns equally well with serial-attention and attention-gradient models of reading.

4. p. 8, 161-162: If I’ve understood this correctly, the flicker effect on pre-targets was observed earlier for low- than high-frequency targets: 72 ms vs. 116 ms after the onset of the pre-target fixation, respectively. This is odd because lexical processing should proceed more rapidly for high- vs. low-frequency words, irrespectively of whether they are located in the fovea or parafovea. More generally, the finding that the flicker effect occurs fairly late in the pre-target fixation but prior to the onset of the target fixation aligns very well with assumptions of serial-attention models (Reichle & Reingold, 2013) and simulations using E-Z Reader (see aforementioned references).

5. Related to concern #4 above, were properties other than word length, frequency, predictability, and plausibility controlled between high- vs. low-frequency target words? For example, the letter trigram “ltz” in the low-frequency target “waltz” is rare and thus likely to be visually salient. I realize that this is only one example (and that it’s an example from the 1st sentence set, which produced null findings), but such information (e.g., bi- and tri-gram frequencies) should probably be provided (e.g., in the Supplemental Materials).

Minor Concerns: These suggestions are admittedly nitpicky but are intended to be helpful.

1. p. 3, line 57: Change “...provide neural evidence for parallel models...” to “...provide evidence consistent with parallel models...”

2. p. 5, line 105: Change “...letters to the word centre...” to “...letters to the left of word centre...”

3. pp. 9-10: The title of the manuscript suggests that individual differences in reading speed will play a significant role in the manuscript. Given that they don’t, I’d suggest changing the title to more accurately reflect the contents of the manuscript.

4. p. 11, line 209: Change “... studies, we ...” to “... studies. We ...”.

Signed: Erik D. Reichle

Reviewer #4 (Remarks to the Author):

- What are the noteworthy results?

The authors present results confirming previously observed eye tracking data but not the theory, thanks to complementary MEG analysis relying on an innovative experimental method to study reading. The article itself is very well written and comprehensible, with detailed figures (rain plots, etc). This is a great study.

- Will the work be of significance to the field and related fields? How does it compare to the established literature? If the work is not original, please provide relevant references.

The work is innovative in terms of experimental design and the analyses of high quality. Results are very relevant to theories about reading and therefore application to dyslexia.

- Does the work support the conclusions and claims, or is additional evidence needed?

Beside a few clarifications, the discussion aligns well with the results. Of potential interest, in Neuroimage 38 (2007) <https://pubmed.ncbi.nlm.nih.gov/17851091/> we used parafoveal priming - and have only behavioural change for short prime to target delay (50ms) suggesting a short lived effect, while MEG data also show a RVF/left hemisphere effect (affecting target 160ms +) suggesting a rapid integration of the parafoveal preview in ongoing target processing -- I was wondering how this fits with your data? not that's I'm forcing you to ref our paper, if you don't think that's relevant to current discussion that's ok.

- Are there any flaws in the data analysis, interpretation and conclusions? - Do these prohibit publication or require revision?

No

- Is the methodology sound? Does the work meet the expected standards in your field?

Yes

- Is there enough detail provided in the methods for the work to be reproduced?

No

A table for material/ items with item values (length, frequency, etc) is common in linguistic studies and should be provided.

Rapid invisible frequency tagging captures lexical parafoveal previewing: in this section, one needs to report some quantification and stats per subjects – since the analysis is done separately for each sensor we have a multiple comparisons issue -- now I'm not saying this is totally wrong. You could report these results in a table per subject (how many sensors show an effect, the mean coherence values, etc) but you should also report an analysis that accounts for multiple comparisons, for instance use the spatial-temporal clustering which tells you which group of sensors show the effect, while accounting for all tests.

Monte-Carlo is the general resampling approach, what FieldTrip does is a permutation – but you need to report what was actually computed: my understanding is that the z-scored mean difference in coherence between baseline and pre-targets is computed, then permuting baseline/pre-target to assess significance (note this is still parametric - the parametric being the mean ; non parametric would be rank based) – this should be rephrased accordingly.

Lexical parafoveal previewing facilitates reading: there is no method description here ; I imagine this is simply the mean speed per subjects vs mean coherence. To strengthen this claim, could the correlation done per sentence/item? (use a HLM to derive the mean correlation per subject and assess if that mean differs from the null)

Data availability: given nowadays standards this seems largely inadequate. The material with details of items and matlab PPT script can be made available in any repository ; this would also greatly

impact reuse of the RIFT approach. Similarly, csv files for data and R script in plots in fig 2, 4, 5 could be shared, allowing future quantitative comparisons.

Sharing of MEG data can be more difficult and you may have ethical constraints too, all this should be explained in the statement. If shared upon request, which data structure will be used? BIDS? how will you ensure people can reuse the data, is it otherwise documented and already archived. Which conditions apply to sharing? Dr Pan is currently a post-doc, what will happen when/if she moves away – who will be able to provide access to the data?

Dr Cyril Pernet

Reviewer #1:

The manuscript reports neural evidence in support of ongoing lexical parafoveal previewing during reading—using an innovative experimental approach combining RIFT, MEG, and eye-tracking. The writing is clear and succinct, analyses are sound, and conclusions are generally reasonable. I congratulate the authors for this excellent work which adds important new knowledge to our understanding of reading, one of the most critical human skills in modern society.

We thank the Reviewer for these positive comments and for their constructive questions that helped us further improve the manuscript.

1. (line 39) Parallel graded attention models. Besides (14, 15) there is another relatively recent model (OB1 reader, Snell et al., 2018), which the author might consider to cite in addition.

We thank the Reviewer for this highly relevant citation. This paper is now cited in the revised manuscript (P.2 with tracked changes):

“In contrast, parallel graded processing models assume that attention is allocated to several words within a reader’s perceptual span in a graded way^{16,17} (for a recent model see the OB1- reader¹⁸)”.

2. (line 43) The statement that impact of the lexical frequency of the upcoming parafoveal word (parafoveal-on-fovea/POF effect) has not been observed is generally correct. However, eye movement recording still represents a powerful tool for evaluation of models. For example, while Ref. (20) does not present evidence for a POF effect, the authors argue that there is a delayed fixation duration effect observable after the boundary. Thus, there are effects that cannot be squared with the strict serial attention model. The bottom line from this comment is that eye movements lend support for parafoveal lexical processing if interpreted with care.

We agree with the Reviewer that some eye movement studies found delayed parafoveal previewing effect, which do not fit the serial attention model. We mention this in the revised manuscript (P.2, with tracked changes):

“However, while eye-tracker studies have been hugely informative, the technique only indirectly measures parafoveal processing. For instance, a few studies applied the gaze-contingent boundary paradigm and found a delayed parafoveal-on-foveal effect, where the fixation durations for word $n+1$ were modulated by the preview difficulty of word $n+2^{22-24}$.”

3. (line 99) Mislocated fixations have been put forward as an explanation for apparent POF effects in the strict serial attention model, however, this cannot explain the effect because of the so-called inverted-optimal viewing position effect of fixation duration (Vitu et al., 2001; Nuthmann et al., 2005).

We thank the Reviewer for this comment, which prompted us to remove the *landing position for pre-target words* analysis (Fig.2b in the old manuscript).

Initially, we intended to use this analysis to show that there are no robust mislocated fixations towards pre-target words. However, this issue has already been addressed by the lack of target lexical effects on pre-target fixations.

4. (line 206) It should also be mentioned that there are syntactic effects from the parafovea (Snell et al., 2017), which might be even more higher-level language processing based on parafoveal information. Also, it should be mentioned that there is evidence for semantic parafoveal-on-fovea effects in Chinese reading (Yan et al., 2009).

We thank the Reviewer for recommending these interesting papers, which are cited in the revised manuscript with tracked changes (P.14):

“It would be of great interest to use the RIFT approach to investigate whether previewing might also occur at higher levels such as extracting semantic information as suggested by natural reading studies of Chinese⁵⁸ and German⁵⁹. Similarly, parafoveal processing at the syntactic level⁶⁰ would also be of great interest to investigate.”

Refs.

Snell, J., van Leipsig, S., Grainger, J., & Meeter, M. (2018). OB1-reader: A model of word recognition and eye movements in text reading. *Psychological Review*, 125(6), 969-984.
<http://dx.doi.org/10.1037/rev0000119>

Yan, M., Richter, E. M., Shu, H., & Kliegl, R. (2009). Readers of Chinese extract semantic information from parafoveal words. *Psychonomic Bulletin & Review*, 16(3), 561-566.
<https://doi.org/10.3758/PBR.16.3.561>

Vitu, F., McConkie, G. W., Kerr, P., & O'Regan, J. K. (2001). Fixation location effects on fixation durations during reading: An inverted optimal viewing position effect. *Vision Research*, 41(25-26), 3513-3533. [https://doi.org/10.1016/S0042-6989\(01\)00166-3](https://doi.org/10.1016/S0042-6989(01)00166-3)

Nuthmann, A., Engbert, R., & Kliegl, R. (2005). Mislocated fixations during reading and the inverted optimal viewing position effect. *Vision Research*, 45(17), 2201-2217.
<https://doi.org/10.1016/j.visres.2005.02.014>

Snell, J., Meeter, M., & Grainger, J. (2017). Evidence for simultaneous syntactic processing of multiple words during reading. *PloS one*, 12(3), e0173720.
<https://doi.org/10.1371/journal.pone.0173720>

Reviewer #2:

This MEG study investigated parafoveal preview effects in sentence reading using a novel Rapid Invisible Frequency Tagging (RIFT) paradigm. The target word in a visually presented sentence was flickered at a frequency of 60Hz. The eye movements during natural sentence reading were monitored using eye-tracking. The coherence between the flicker signal and the brain response during the fixation of the pre-target word was the main dependent variable. Its modulation by the lexical frequency of the target word is interpreted as evidence that readers access lexical information of the upcoming word parafoveally.

This study addresses a long-standing question in psycholinguistic reading research using a novel and innovative experimental paradigm. Previous research has shown that lexical parafoveal preview effects on behaviour and brain responses, if they exist at all, are subtle at best. The current study adds new empirical data to this debate. The authors are correct in saying that previous research has failed to provide conclusive results with regard to parafoveal lexical processing. We think that the authors' approach provides an interesting new angle on this issue, but as the effects are subtle and several aspects of this study are unclear to us, we are not yet convinced that this new evidence is conclusive either.

We thank the Reviewer for these comments and for the interesting suggestions that helped us further improve the manuscript.

1) The authors discuss the fact that they found a subtle preview effect on coherence values but not on behaviour (ll. 219ff., "One might ask why..."). This raises the question whether the observed effects, if reliable, really reflect the "simultaneous processing of several words" in a way that is relevant for the cited computational models, or a separate "detection process", which may not be relevant for reading behaviour (and therefore not within the scope of those models, but possibly interesting for other reasons). The authors attempt an explanation in terms of "attracting more or less attention", but if this is a valid explanation (which we think was also the prediction in previous behavioural studies on this topic) then one would indeed expect behavioural effects?

We thank the Reviewer for this comment as it helped us clarify the points that were not sufficiently clear in the original version of our manuscript.

First we would like to stress that we do find behavioural evidence for lexical previewing as evidenced by the correlation between neuronal effects and reading speed. We have now elaborated on this finding (P12 in the revised manuscript with tracked changes):

"One might ask why lexical parafoveal previewing is reflected in neuronal responses (Fig. 4b) but not in fixation durations (Fig. 2). We would like to stress that albeit lexical parafoveal processing was not reflected in the pre-target fixation durations, the neuronal effects were linked to reading speed. Basically participants who read faster also have a stronger parafoveal lexical neuronal modulation. This suggests that parafoveal processing is reflected by the allocation of covert attention, especially to less common targets words. However, this allocation of covert attention does not directly impact overt attention, i.e. the decision criteria for when to initiate the saccade. Possibly, the absence of lexical previewing effects on pre-target fixation times might facilitate fluent reading. Prolonging the current fixation when previewing a difficult word is not an efficient strategy, since it means keeping the difficult word in the low visual acuity parafovea for longer. Taken together, our study shows that

natural reading involves the simultaneous processing of several words in a graded way, providing novel neural evidence for the idea that “readers are parallel processors”⁴¹

Maybe the RIFT is more sensitive than behaviour. Is this something that could be analysed in more detail, e.g. in terms of power analysis and comparing number of stimuli and participants with previous studies?

Good point. For the statistical power, we have estimated Cohen’s d value to show the effect size for t-tests of behavioural and neural data. In Line 96, Cohen’s d value for the difference in fixation duration of pre-target words with respect to lexical frequency is 0.03, indicating that the behavioural effect is indeed very small. In Line 157, Cohen’s d value for pre-target coherence is 0.43, indicating a medium effect size for neural data. In order to justify the null finding in behaviour data, Bayes Factor analysis was also performed. The Bayes factor is 0.175, which favours null hypothesis not the alternative hypothesis. We believe this power analysis qualify the sensitivity of both behavioural and neural data.

For the concern about number of stimuli and participants: we have 228 sentences from two sentence sets in the current study, which is higher compared with previous eye-tracking studies. In order to replicate our main result (Fig. 4), we redid the same analysis for the sentence set that has been published (Degno et al., 2019). Degno et al. used 108 sentences with 42 participants, and they found no lexical parafoveal previewing on neither behavioural nor neural data (fixation-related potentials analysis, we will also show this analysis in reply to your 4th comment). In the current analysis, we only used 86 sentences from Degno et al. (after removing the sentences that contain the same target words as in our own sentence set) with 26 participants (tagging responders). The lexical parafoveal previewing effect is shown below:

Figure R1. Pre-target coherence over the tagging response sensors for the sentence set in Degno et al. (2019). The parameters and procedure for this coherence analysis were exactly the same as in Fig. 4 (main text).

In short, we replicated our main result with a relatively reduced number of sentences (86) and participants (26) compared with Degno et al. 2019 (108 sentences and 42 participants). Thus, we believe that the tagging response measured by coherence is stable and sensitive enough to capture the lexical parafoveal previewing.

2) Coherence for the pre-target fixation was stronger for low compared to high frequency target words. This is not discussed, even though it may support the “attention explanation” of the lexical

preview effects: low frequency words are harder to process and require more attention. On the other hand, high frequency words may grab more attention (in this case for example in order to trigger an earlier saccade). This would fit better with the attention literature, where more salient and more familiar stimuli are usually easier to detect and more distracting. This would deserve a more detailed discussion. And again, these explanations would also lead to predictions for behavioural effects.

We thank the Reviewer for this comment, which indeed raises an interesting question.

We argue that the pre-target coherence is higher for low frequency targets because they are more difficult to process and require more attention. The lexical previewing effect (pre-target coherence difference) is observed in visual cortex, but it is likely due to top-down modulation from areas like VWFA.

We add a more detailed discussion about this in the revised manuscript with tracked changes (P.13):

“We interpret the stronger frequency tagging for low-frequency target words as being a consequence of the allocation of more covert attention to the parafoveal word. The covert attention will help to facilitate the processing of less familiar words. Alternatively, one might have expected high-frequency words to be more attention grabbing as they are more familiar; however, the frequency tagging result suggest that this is not the case.”

3) The authors provide an explanation for the absence of tagging effects for target word fixations (l. 179). Does this mean this paradigm can only be used to study parafoveal vision? If we understood correctly, at least the cited Drijvers et al. HBM 2020 study used frequency tagging for foveal stimuli.

We thank the Reviewer for this point, which prompted us to explain our rationale in more detail. We added a new paragraph as below (P.13 with tracked changes in the revised manuscript):

“Why did we not find differences in neuronal excitability with respect to lexical effects for the foveal target words, even though previous studies have shown that RIFT can be used to investigate both foveal^{27,28} and parafoveal processing^{25,26}? In these previous studies, the flickering stimuli are larger (at least 5.7° visual angle in diameter) compared to the size of the flickering words used in this study (at the most 3° by 1° visual angle). Also the durations to perceive the flicker were longer (at least 1.5s) compared with here (around 0.2s). In addition, flicker sensitive photoreceptors (rod cells) are more abundant in the parafovea of the retina⁴². As a consequence, there would be higher sensitivity to flickering in the parafovea compared to the fovea.”

4) These results would be much stronger if the authors could replicate previously reported word frequency effects for target word fixations on EEG/MEG data in a conventional evoked analysis.

We thank the Reviewer for this suggestion, which we followed (P.10 in the revised manuscript with tracked changes):

“Late lexical parafoveal effect observed from fixation-related fields

In order to investigate if the lexical previewing effect could also be observed in fixation-related fields (FRFs), we compared FRFs for pre-target words followed by low and high lexical frequency target words over all participants (n = 39).

We conducted a cluster-based permutation test over all combined planar gradiometers (0 – 0.5 s, aligning with fixation onset for pre-target words). We found clusters of sensors that had significantly higher FRFs when the pre-target word was followed by low compared with high lexical target words

(Fig. 6a, $P_{cluster} < 0.05$, two-tailed pairwise t-test, 1000 permutations). Averaged pre-target FRFs over these significant sensors are shown in Fig. 6b. We observed the strongest effect around 0.4s in the left posterior sensors. We formed the same FRFs analysis for target fixations but we did not find a significant difference (see Supplementary Figure 4 for the target averaged FRFs over sensors shown in Fig. 6a).

Fig. 6. Fixation-related fields for pre-target words. (a) Topography for the sensors that showed a lexical parafoveal previewing effect during pre-target fixations. The colour bar indicates the FRFs difference for pre-target words followed by low compared with high lexical target words ($n = 39$, cluster-based permutation with $p < 0.05$, two-tailed pairwise t-test). (b). Pre-target FRFs over these significant sensors for low (blue) and high (orange) lexical target conditions. The shaded area indicates standard error over 39 participants.”

We performed the same FRFs analysis for target words, but no significant lexical effect was found. It is in the supplementary material now copied as below (P.6 with tracked changes):

“Supplementary Figure 4. Fixation-related fields (FRFs) for target words. We performed the same FRFs analysis for target fixations as in Fig. 6, but found no significant difference with respect to lexical frequency ($n = 39$, cluster-based permutation with $p < 0.05$, two-tailed pairwise t-test, 1000 permutations). Here we show the averaged FRFs for low (blue) and high (orange) lexical target words over the sensors shown in Fig. 6a. The shaded area indicates standard error over 39 participants.”

In the discussion we now write (P.14 in the revised manuscript with tracked changes):

“Some studies did not find evidence for lexical previewing in the FRPs^{20,21}, while one other study³⁰ found this effect around 400 ms, compatible with our findings (see Fig. 6b). This late effect probably reflects the integration of target words into prior context. In comparison, the RIFT approach allowed us to capture parafoveal previewing at a much earlier stage of word processing. As such the FRFs/FRPs and the RIFT approach provide complementary information about word processing during natural reading.”

The related method section is copied as below (P. 25 in the revised manuscript with tracked changes):

“Fixation-related fields analysis

This analysis was performed for all participants (n = 39). For the 1s pre-target epochs (-0.5 to 0.5 s with fixation onset to the pre-target word), we applied a 35 Hz low-pass filter using phase preserving two-pass Butterworth filters. For each participant, the same number of trials were randomly selected based on the minimum trial number across the low- and high- frequency conditions. After grand averaging the epochs with a baseline subtraction (-0.2 to 0 s), we obtained fixation-related fields (FRFs) for pre-targets in both conditions. For each pair of planar sensors at the same location, we combined the root-mean-square of the FRPs.

A permutation test was conducted for the averaged pre-target FRFs over all valid sensors to compare the lexical effects. A cluster-based permutation test was conducted for the whole time window (0 to 0.5s) with a threshold of $p = 0.05$ (two-tailed pair-wise t-test). To form a cluster required at least two neighbouring sensors that exceeded the threshold. The neighbourhood layout for sensors was pre-defined by FieldTrip for the MEGIN system. Next, we created a reference distribution by permuting the data 1000 times. In each permutation, the labels for low and high target conditions were randomly assigned over participants, and the t-values were calculated to obtain the clusters. The maximum summed t-value from each cluster was used to construct the reference distribution. After 1000 permutations, we sorted all t-values in the null distribution from the minimum to the maximum. T-values at the 25th and the 975th position were chosen as the thresholds for negative ($FRFs_high > FRFs_low$) and positive ($FRFs_low > FRFs_high$) clusters in the experimental data. We only found significant positive clusters ($p < 0.05$), indicating that FRFs were higher for pre-target words followed by low compared with high lexical target words.”

5) The latency analysis of the preview effect is interesting (ll. 154ff.). However, an effect is first “observed” but then only found “robust” in one dataset, which is attributed to word length effects. The word length hypothesis could be tested as an interaction with word length. The presence of an effect in only one dataset, without further justification or an interaction, is not strong evidence for anything. This analysis could be described as exploratory, and possibly moved to the supplementary section.

We thank the Reviewer for this comment as it helped us to make the manuscript more succinct. We deleted the latency analysis from the main text and only mention it briefly in the new manuscript as below (P.8 with tracked changes):

“Moreover, this lexical parafoveal previewing also modulated pre-target coherence onset latency when both pre-target and target words were short (see Supplementary Figure 5).”

6) A more complete set of psycholinguistic variables for the target and pre-target words must be provided. Table 1 does not show ‘length’ and ‘position’ for low and high frequency words separately, and it does not show important orthographic variables such as bi/trigram frequency and Coltheart’s

neighbourhood size or orthographic Levenshtein distance, which are important orthographic confounds for possible preview effects.

We thank the Reviewer for this comment and agree that it is important.

First, we revised Table 1 by adding details about the length and position as below (P.4 in the new manuscript with tracked changes):

“Table 1. Characteristics of words used in the current study.

	Pre-target	Low freq target	High freq target	Post-target
Word frequency	359.9 (1109.3)	5.3 (4.5)	95.3 (135.5)	569.3 (1734.7)
Word Length	6.1 (1.5)	5.8 (0.8)	5.8 (0.8)	6.7 (1.7)
Position	5.7 (2.3)	6.7 (2.3)	6.7 (2.3)	7.7 (2.3)

Note. All values here are mean values, standard deviations are in parentheses. Low (<10) and high lexical frequency (>30) target words are reported in terms of the total CELEX frequency per million²⁹. Word length is the number of letters in a word. Position refers to the location in a sentence where the target word is presented and is measured in the number of words. The sentences were 11.2 ± 2.1 words long.”

We made a new table for the orthographic variables for the target words, reported in the supplementary material as below (P.1 with tracked changes):

“Supplementary table. Orthographic variables for the target words

	Low lexical freq target	High lexical freq target	t values
Bigram type freq	54.0 (41.2)	61.1 (46.2)	-1.741 (p = 0.083)
Bigram token freq	979.2 (737.2)	1332.7 (886.7)	-4.816 (p = 2.6e-06)
Trigram type freq	9.4 (12.2)	11.6 (14.0)	-1.810 (p = 0.072)
Trigram token freq	153.7 (288.0)	320.6 (334.2)	-6.060 (p = 5.2e-09)
Neighborhood size	1.9 (2.4)	2.4 (2.6)	-2.507 (p = 0.013)

*Note. All measures are mean values estimated from the N-watch program¹. The standard deviations are in parentheses. Pair-wised *t*-tests were conducted for each measurement between low and high lexical frequency target words, the *t* values are shown in the third column with the *p* values in parentheses.*

Bigrams refer to two successive letters in a string, e.g., bigrams for the word edge are ed, dg, and ge. For each bigram, the type frequency is the number of all 4-letter words that contain this bigram in the same position – length and position sensitive. E.g., the type frequency for ed is 4 (eddy, edge, edgy, and edit). While the token frequency for each bigram is the sum of the word frequencies for all types. E.g., the token frequency for ed is 81, being the sum of word frequencies for these four words (eddy, edge, edgy, and edit). The bigram type/token frequency for the whole word is the averaged type/token frequencies over all bigrams. Trigram is three successive letters in a string, e.g., trigrams for the word edge are edg and dge. The trigram type and token frequency are calculated in a similar way as for bigrams.

Neighbourhood size of a string indicates how many words can be formed by just substituting one letter in the string."

We conducted a control analysis for the orthographic variables of the target words, and added a paragraph about it in the new manuscript as below (P. 9 with tracked changes):

"No confounding factor from the orthographic information

One potential concern for the current study is the influence of orthographic information. First, we found that bi/trigram type frequency did not co-vary with word frequency, but bi/trigram token frequency and neighbourhood size did (see supplementary table). Next, we qualified a possible orthographic parafoveal previewing effect by separating trials based on these three co-varying orthographic variables. All parameters and procedures were the same as in the lexical previewing analysis (Fig. 4). However, no significance was found for bi/trigram token frequency or neighbourhood size (p-values were 0.69, 0.06, and 0.44 separately, Bonferroni corrected). Please note that trigram token frequency co-varied greatly with word frequency, which could explain the marginal significance. We conclude that orthographic information is not a significant confounding factor for the reported neural effects on lexical parafoveal previewing effects."

Figure R1. Three control analyses for orthographic confounds to the lexical parafoveal previewing effect.

7) Source estimation results are presented for the tagging response, but not for the frequency effect. The authors then interpret the results as if they are specific to visual cortex. This interpretation would be stronger with a separate whole-brain or ROI analysis.

We thank the Reviewer for this point, which helped us to clarify our analyses. In the present study we used pre-target coherence as an index for parafoveal processing. Therefore, we used DICS to localize the source of pre-target coherence difference (low-high). A cluster-based permutation test was conducted to search for the potential effect over the whole-brain, but no robust difference was found. We can see from the figure below that source with the highest coherence difference was localized to the left orbitofrontal (BA11), which is probably due a low signal-to-noise ratio of coherence difference. As such we do not feel comfortable including this analysis.

Figure R2. Source localization for the lexical parafoveal previewing effect over the whole brain.

We clarify the ROI-based analysis in the revised manuscript as below (P.6 with tracked changes):

“Since mainly sensors over visual areas responded to the visual flicker, we first identified the sensors with robust tagging responses. We compared the 60 Hz visual flicker-to-MEG coherence during pre-target fixations (caused by the target flickering in the parafovea) with a baseline period (the cross-fixation presented before sentence onset). Robust tagging responses were found over the left visual cortex sensors (Fig. 3a), reflecting the neural resources associated with parafoveal previewing. This analysis was conducted by pooling data over both target lexical frequency conditions (for details see Methods). In 26 out of the 39 participants, one or more sensors showed significant tagging responses (Fig. 3a, 5.4 ± 4.0 sensors per participant, mean \pm SD; for topography for each participant, see Supplementary Figure 2). The subsequent analyses were based on these participants and sensors.”

8) The main analysis was run on 26 out of 39 participants who showed a significant tagging response (ll. 121ff.). This is fine, but shows that this response is rather subtle. Is this common practice in this kind of experiment? We could not find this kind of information in Drijvers et al. HBM 2020 and Zhigalov et al. NI 2019. Is the absence of tagging responses in one third of the participants assumed to be mostly physiological/anatomical in nature, or could this point to individual differences in reading behaviour? For example, could the non-responders still be included in the correlation analysis?

We thank the Reviewer for this concern, which is important for the validity of the RIFT technique.

First, we would like to argue that almost all participants in our previous RIFT studies showed stable tagging response when the flickering stimulus is big and the flickering duration is long. Following are the typical flickering parameters:

In Drijvers et al. HBM 2020, the flickering rectangle was “10.0 by 6.5° of visual angle (width by height), with a tagging duration of 2s long.

In Zhigalov et al. NI 2019, the rounded flickering stimulus was 5.7° of visual angle in diameter, with a tagging duration of 1.5s long.

However, in the current natural reading paradigm, the flickering target word is 3 by 1° at most, and the time window when the flicker was perceived is around 0.2s (averaged pre-target fixation duration). In addition, participants make saccades continuously. All these factors will make the tagging responses weaker, which resulted in fewer participants being selected. We have noticed after data collection that participants with thick hair (e.g. in a braid) or a comparatively smaller head (resulting in more space between the visual cortex and the MEG helmet) were less likely to produce a stable tagging response.

In order to justify the participant selection procedure, we show the topography of the tagging response for each participant in the supplementary Figure 2 and copy it as below (P.3 in the revised version with tracked changes):

“Supplementary Figure 2 | Topography of significant tagging response sensors for each participant. A Monte-Carlo based permutation test was performed to select sensors that showed stronger coherence to the tagging signal during the pre-target fixation period compared with the baseline period (no flicker). Significant sensors are marked with circles in the topography, with the values indicating coherence difference (pre-target minus baseline). Participants who had significant tagging response sensors are shown with a bold subtitle. Further coherence analyses (Fig.4, Fig.5, and Supplementary Figure 3) were based on the marked sensors.”

9) Methodological clarifications:

We thank the Reviewer for these comments, which helped us to clarify several methodological details.

a. PCA was used to reduce the rank of the data to 30 components (l. 408). Is this a common approach in this field (it also appears to be different from the other two frequency tagging studies)? If the subtle experimental effects are reflected in the lower components then this approach may lose information.

We didn't address the ICA section correctly in the original manuscript, and have now revised this in the new version as below indeed relying on common practice (P21. with tracked changes):

"After removing malfunctioning sensors (0 to 2 sensors per participant), these segments entered an independent component analysis (ICA)⁶⁶. Data were decomposed into independent components with the same number of sensors. Next, the components related to eye blinks, eye movements, and heartbeat were rejected. Finally, we manually inspected all these segments to further identify and remove any segments that were contaminated by excessive noise like ocular, muscle, or movement artefacts."

Since we only use ICA to remove ocular and cardiac related artefacts, we don't think it will affect the tagging responses or any other non-noise signal.

b. Please provide some more details on how the covariance/cross-spectral matrices for the beamforming analysis were computed and regularized. Was the reduction in rank taken into account in the construction of the beamformer?

We added more details to the source estimation as below (P. 24 in the revised manuscript with tracked changes):

"The Cross-Spectral Density (CSD) matrix was calculated at 60 Hz between all possible combinations chosen from the MEG sensors and photodiode (MEG sensor + MEG sensor as well as r MEG sensor + photodiode). This was done for each participant during the 1s long pre-target and baseline segments, after using a hamming taper. No regularisation was performed to the CSD matrices ($\lambda = 0$)."

The data rank reduction was an inaccurate description in the original manuscript, actually no rank reduction was conducted (please see the reply above).

c. The authors use "coherence" and "r²" in different places. This should be consistent, e.g. "coherence" throughout.

We deleted the "r²" and use "coherence" throughout.

We understand the rationale for using coherence. However, can the phase of the tagging response be assumed to be stable at this (relatively high) frequency?

The phase at 60Hz is quite stable as shown in Figure R2 below. We aligned phase series at 60Hz across different epochs in a fixation related fashion (i.e., fixation onset to pre-target words) and obtained the grand average for the tagging signal (green line), pre-target epochs (black line), and no-flicker baseline epochs (grey line). Figure R2 shows that the phase information during pre-target fixations is driven by the tagging signal, while no such phase consistency across trials is observed for the no-flicker baseline (grey line).

We added more details as below (P. 6 in the revised manuscript with tracked changes):

“In addition, coherence is quite stable even at a high-frequency band³¹.”

Figure R4. Consistent phase information driven by tagging signal in the pre-target fixations.

Would it be interesting to run a separate analysis on signal power at the tagging frequency (as in Zhigalov et al. 2019)?

First, we do observe a clear peak around 60Hz in the power spectrum for pre-target fixation as in Zhigalov et al. (2019), as shown below in Supplementary Figure 3 (with tracked changes in the revised Supplementary Material). Next, we selected the tagging response sensors based on 60Hz power, but it is noisier than coherence as shown in the topographies (Figure R5).

We added more details as below (P. 5 in the revised manuscript with tracked changes):

“A measure of time-resolved coherence between the 60 Hz visual flicker and the brain activity was used to quantify the tagging responses (see Methods for details). Since coherence is based on the phase relationship between the photic driving signal and the brain response, it is a more sensitive measure than power of the brain response at the tagging frequency.”

“Supplementary Figure 3 | Power spectrum for the pre-target fixation period over significant tagging response sensors. For each participant ($n = 26$), power was calculated from all tagging response sensors (from 2 to 80 Hz; 1 Hz steps; Hanning taper). This was conducted for all pre-target epochs (indicated by the black line), pre-target words that were followed by low lexical frequency targets (blue line), pre-target words that were followed by high lexical frequency targets (orange line), and the baseline epochs (grey line). Next, power values in each condition were averaged across all tagging response sensors and then averaged across all participants. These power values were then transformed as the relative change with baseline power to get rid of the $1/f$ component (at each frequency point, $(Pow - Pow_{baseline})/Pow_{baseline}$). We can see a clear peak around the tagging frequency at 60 Hz for both pre-target word conditions but not the baseline condition, indicating a stable tagging responses to the flickering target words in the parafovea.”

Figure R5. Topographies for tagging response sensors selected by 60Hz power (a) and 60Hz coherence (b, the same figure as shown in Fig. 3).

d. It wasn't clear to us which exact latency ranges were used for the statistical frequency tagging analysis (ll. 144ff.).

We now clarify the statistical time window selection in the new manuscript and copy it as below (P.8 with tracked changes):

“To ensure that the coherence in the pre-target fixation intervals was not contaminated by temporal smoothing from the target fixation, the time window for averaging coherence was adjusted individually according to the shortest pre-target fixation duration. For each participant, we identified the shortest duration (t) over all pre-target fixations (t was 88.3 ± 8.9 ms across participants, mean \pm SD). Next, pre-target coherence for the two conditions was averaged within this time window, then averaged over the sensors of interest to obtain the grand average. Because the number of trials biases the magnitude of the coherence measure, we subsampled the same number of trials for both conditions in each participant (by randomly selecting trials from the condition with more trials). The difference in coherence across participants was compared using a paired t -test demonstrating that the flicker response was significantly stronger for target words in the parafovea with a low compared to high lexical frequency (Fig. 4b, $t(25) = 2.20$, $p = 0.037$, $d = 0.43$, two-tailed pairwise t -test).”

Reviewer #3:

Summary. The manuscript describes an eye-tracking experiment in which high- vs. low-frequency targets words in the parafoveal were “flickered” at 60 Hz (using RIFT) to determine if the neuronal activity induced by this manipulation could be measured (using MEG) from the pre-target word. This manipulation did produce a frequency-modulated neuronal response for short words (from one of two sets of sentences), but interestingly, target-word frequency did not modulate fixations on the pre-target words (i.e., there was no evidence for parafoveal-on-foveal effects). The former effect was also correlated with reading speed; as reading speed increased, the size of the MEG-measured neuronal activity increased. On the bases of these findings, the authors argue that their study is more consistent with attention-gradient models of reading, wherein multiple words are subject to concurrent lexical processing, rather than serial-attention models, wherein only one word is subject to lexical processing at any given time.

General comments. I really enjoyed reading this manuscript. The manuscript itself was well written, and the experiment was innovative and appeared to be methodologically sound. (Please note, though, that I'm not an expert on RIFT or MEG). Possibly offsetting these strengths, however, is the fact that the experimental results are open to an alternative interpretation—one that is actually consistent with the serial-attention models that the authors argue against. My concerns about the manuscript (both major and minor) are outlined below.

We thank the Reviewer for these positive and helpful comments and hope that we were able to address the remaining concerns.

Major Concerns/Questions:

1. p. 2, lines 34-35: The authors claim that, according to serial-attention models, “lexical information is not generally extracted from the parafovea.” This statement is incorrect. For example, according to the E-Z Reader model, most word skipping reflects instances in which the parafoveal word has received sufficient lexical processing to cancel the saccade that would otherwise cause the word to be fixated (Reichle & Drieghe, 2015). Similarly, simulations using E-Z Reader (Schotter, Reichle, & Rayner, 2014) indicate that, within the framework of a strictly serial-attention model, significant time is available for late lexical (e.g., semantic) processing of upcoming words.

We thank the Reviewer for this point which has helped us to develop a more comprehensive description about the E-Z Reader model. We agree that the statement needed more nuance and we have now changed this (P.2 in the new manuscript with tracked changes):

“Serial attention shift models maintain that lexical processing is restricted to one word at a time⁸⁻¹³, but that attention can be shifted to the next word before the eyes do, allowing significant parafoveal processing¹⁴. According to the mechanism described in the E-Z Reader model, the parafoveal processing can explain, for example, word skipping effects¹⁵”

2. p. 5. The authors report a fixation landing-position distribution (Fig. 2b) to argue against the notion that their observed effect of parafoveal flicker was due to mis-located fixations (i.e., pre-target fixations intended for the targets but that fell short due to saccadic error). Unfortunately, this analysis fails to rule out the possibility of mis-located fixations: Because each letter subtended 0.35 visual degrees (p. 15) but calibration error is up to 1 visual degree (p. 18), I think it’s reasonable to assume that some proportion of the pre-target fixations were intended for the targets, suggesting that overt attention (as measured by the flicker manipulation) may have already been focused on the targets. Simulations have demonstrated how purported parafoveal-on-foveal effects can arise due to saccadic error and/or eye-tracking measurement error even in a strictly serial-attention model (Reichle & Drieghe, 2015).

We thank the Reviewer for this comment, which prompted us to remove the landing position segment for the pre-target words analysis (Fig.2b in the old manuscript). Please note that mis-located fixations have been proposed to argue against parafoveal-on-foveal *reading time* differences, which we did not find (see also our reply to Reviewer 2 regarding landing position). In addition, and interestingly, we did not observe a coherence effect on the target itself. Hence, if these mis-located fixations were due to calibration error, then they should not have contributed to the pre-target coherence effect that we observed.

3. p. 6, line 129: The source localization analyses suggest that the flicker effect was “localized in the early visual cortex.” Later in the manuscript (p. 12), the authors acknowledge that this is at odds with what might have been predicted (e.g., word-frequency effects are generally localized in the visual word-form area), but then argue that interactive models might account for this finding via “top-down modulation ... over sensory cortices.” This account seems ad hoc and lacks parsimony. A more parsimonious account might simply be that the experiment was successful at demonstrating concurrent visual processing of foveal and parafoveal words—a finding that aligns equally well with serial-attention and attention-gradient models of reading.

We thank the Reviewer for the suggestion but would like to maintain our argument that the tagging modulation in early visual cortex is a consequence of attention allocation. We have now expanded the rationale in the new manuscript and copy it as below (P.13 with tracked changes):

“The neuronal response reflecting previewing was observed in the early visual cortex. This might be a surprise, as functional Magnetic Resonance Imaging studies have localized lexical frequency to e.g. the visual word form area⁴³. According to interactive processing theories, higher-level lexical information interacts with lower-level visual information during word recognition^{38,44} and the feedback modulation can be measured by MEG over sensory cortices⁴⁵. Thus, lexical frequency information extracted in the parafovea could direct visual attention covertly. Increased spatial attention will boost tagging responses^{25,26}, resulting in stronger coherence for the pre-target word followed by a low compared with a high lexical frequency target word. We interpret the stronger frequency tagging for low-frequency target words as being a consequence of the allocation of more covert attention to the parafoveal word. The covert attention will help to facilitate the processing of less familiar words. Alternatively, one might have expected high-frequency words to be more attention grabbing as they are more familiar; however, the frequency tagging result suggest that this is not the case.”

4. p. 8, 161-162: If I’ve understood this correctly, the flicker effect on pre-targets was observed earlier for low- than high-frequency targets: 72 ms vs. 116 ms after the onset of the pre-target fixation, respectively. This is odd because lexical processing should proceed more rapidly for high- vs. low-frequency words, irrespectively of whether they are located in the fovea or parafovea. More generally, the finding that the flicker effect occurs fairly late in the pre-target fixation but prior to the onset of the target fixation aligns very well with assumptions of serial-attention models (Reichle & Reingold, 2013) and simulations using E-Z Reader (see aforementioned references).

We thank the Reviewer for this point and agree that a high frequency word should be recognized earlier than a low frequency word in foveal vision, as supported by the lexical effect observed in target fixation durations. The coherence onset latency was an exploratory analysis, which was only significant for one subset of sentences. We have now moved this part to the Supplementary material as suggested by Reviewer 2 in point 5. We only mention it briefly in the new manuscript as below (P.8 with tracked changes):

“Moreover, this lexical parafoveal previewing also modulated pre-target coherence onset latency when both pre-target and target words were short (see Supplementary Figure 5).”

We are, however, somewhat confused by the Reviewer’s description of this effect as a “fairly late” effect. According to Reichle and Reingold (2013), in a serial model that allows for substantial parafoveal processing, parafoveal lexical processing only starts at about 150ms post pre-target fixation onset. (These values are in line with Schotter, Reichle, and Rayner’s, 2014, simulations of a 98ms preview time – 240ms-98ms=142ms – and Reingold et al.’s, 2012, survival-curve analyses.) The fact that we already found a frequency effect well before attention has purportedly shifted to the next word, indicates that a substantial amount of lexical parafoveal processing must have occurred much earlier.

We acknowledge the issue mentioned by the Reviewer and addressed it in the Supplementary note (P. 7 in the revised Supplementary Material with tracked changes)

“In particular we find that the neuronal responses for the low-frequency words tend to occur earlier than for high frequency words. This might seem at odds with the expectation that lexical processing

is faster for high than low frequency words. However, the exact onset of the neuronal response should be interpreted with caution, as it is co-modulated by the magnitude of the response. "

5. Related to concern #4 above, were properties other than word length, frequency, predictability, and plausibility controlled between high- vs. low-frequency target words? For example, the letter trigram "ltz" in the low-frequency target "waltz" is rare and thus likely to be visually salient. I realize that this is only one example (and that it's an example from the 1st sentence set, which produced null findings), but such information (e.g., bi- and tri-gram frequencies) should probably be provided (e.g., in the Supplemental Materials).

We thank the Reviewer for this comment on possible confounds, which was also mentioned by the 2nd Reviewer. Please see the response to point 6 of Reviewer 2 (page 7 in this response letter).

Minor Concerns: These suggestions are admittedly nitpicky but are intended to be helpful.

We thank the Reviewer for all these detailed suggestions, which we have now incorporated in the new version of our manuscript with tracked changes.

1. p. 3, line 57: Change "...provide neural evidence for parallel models..." to "...provide evidence consistent with parallel models..."

Done.

2. p. 5, line 105: Change "...letters to the word centre..." to "...letters to the left of word centre..."

This paragraph is related to the landing position analysis that has been deleted in the new manuscript.

3. pp. 9-10: The title of the manuscript suggests that individual differences in reading speed will play a significant role in the manuscript. Given that they don't, I'd suggest changing the title to more accurately reflect the contents of the manuscript.

We thank the Reviewer for this concern. However, the correlation result provides a behavioural benefit for the lexical parafoveal previewing, which we think might be of interest to a large proportion of readers. We would therefore like to keep the original title.

4. p. 11, line 209: Change "... studies, we ..." to "... studies. We ...".

Done.

Reviewer #4:

- What are the noteworthy results?

The authors present results confirming previously observed eye tracking data but not the theory, thanks to complementary MEG analysis relying on an innovative experimental method to study

reading. The article itself is very well written and comprehensible, with detailed figures (rain plots, etc). This is a great study.

We thank the Reviewer for these positive comments.

- Will the work be of significance to the field and related fields? How does it compare to the established literature? If the work is not original, please provide relevant references.

The work is innovative in terms of experimental design and the analyses of high quality. Results are very relevant to theories about reading and therefore application to dyslexia.

We really appreciate these positive comments.

- Does the work support the conclusions and claims, or is additional evidence needed?

Beside a few clarifications, the discussion aligns well with the results. Of potential interest, in Neuroimage 38 (2007) <https://pubmed.ncbi.nlm.nih.gov/17851091/> we used parafoveal priming - and have only behavioural change for short prime to target delay (50ms) suggesting a short lived effect, while MEG data also show a RVF/left hemisphere effect (affecting target 160ms +) suggesting a rapid integration of the parafoveal preview in ongoing target processing -- I was wondering how this fits with your data? not that's I'm forcing you to ref our paper, if you don't think that's relevant to current discussion that's ok.

We thank the Reviewer for this interesting paper, which shares some common ideas with our study. In the NeuroImage paper, the reaction time for the target word was significantly shorter for the congruent prime condition, indicating that valid parafoveal information facilitated lexical access for the target word. A similar effect is shown in our data as the parafoveal lexical processing facilitated reading speed. We believe that the neural data point to the importance of using a natural reading paradigm, since the neural processes for words are different given the availability of parafoveal information.

We now cite this paper in the new manuscript as below (P.14 with tracked changes):

"..., addressing the importance of using natural reading paradigms. The importance of natural reading paradigms is also supported by an MEG study that found different neural patterns when the priming word was in the fovea and parafovea, where the latter is relevant to natural reading⁵⁷."

- Are there any flaws in the data analysis, interpretation and conclusions? - Do these prohibit publication or require revision?

No

- Is the methodology sound? Does the work meet the expected standards in your field?

Yes

- Is there enough detail provided in the methods for the work to be reproduced?

No

1. A table for material/ items with item values (length, frequency, etc) is common in linguistic studies and should be provided.

We thank the Reviewer for this comment which helped us to give a clearer description of the material.

We revised Table 1 by adding details about the length and position as below (P.4 in the new manuscript with tracked changes):

“Table 1. Characteristics of words used in the current study.

	Pre-target	Low freq target	High freq target	Post-target
Word frequency	359.9 (1109.3)	5.3 (4.5)	95.3 (135.5)	569.3 (1734.7)
Word Length	6.1 (1.5)	5.8 (0.8)	5.8 (0.8)	6.7 (1.7)
Position	5.7 (2.3)	6.7 (2.3)	6.7 (2.3)	7.7 (2.3)

Note. All values here are mean values, standard deviations are in parentheses. Low (<10) and high lexical frequency (>30) target words are reported in terms of the total CELEX frequency per million²⁸. Word length is the number of letters in a word. Position refers to the location in a sentence where the target word is presented and is measured in the number of words. The sentences were 11.2 ± 2.1 words long.”

We also included all the sentences that were used in the current study in the Appendix of Supplementary material (P. 12 with tracked changes). For the interest of brevity, we don't copy the Appendix here. We mentioned this in the Stimuli section in revised main text as below (P. 16 with tracked changes):

“For the full list of all the sentences that used in this study, please see Appendix in the Supplementary material.”

2. Rapid invisible frequency tagging captures lexical parafoveal previewing: in this section, one needs to report some quantification and stats per subjects – since the analysis is done separately for each sensor we have a multiple comparisons issue -- now I'm not saying this is totally wrong. You could report these results in a table per subject (how many sensors show an effect, the mean coherence values, etc) but you should also report an analysis that accounts for multiple comparisons, for instance use the spatial-temporal clustering which tells you which group of sensors show the effect, while accounting for all tests.

We thank the Reviewer for this point which helped us to clarify the rationale of the analyses. Because our previous studies showed that not all brain regions have a stable response to the visual flicker, we first ran a selection procedure to find the sensors of interest. We clarify this procedure in the revised manuscript as below (P.6 with tracked changes):

“Since mainly sensors over visual areas responded to the visual flicker, we first identified the sensors with robust tagging responses. We compared the 60 Hz visual flicker-to-MEG coherence during pre-target fixations (caused by the target flickering in the parafovea) with a baseline period (the cross-fixation presented before sentence onset). Robust tagging responses were found over the left visual cortex sensors (Fig. 3a), reflecting the neural resources associated with parafoveal previewing. This analysis was conducted by pooling data over both target lexical frequency conditions (for details see Methods).”

Since this sensor selection procedure is orthogonal to the lexical parafoveal effect analysis, we circumvent the multiple comparison problem.

We agree with the Reviewer that individual statistics for the tagging response is relevant, and we now show the topography of each participant in the supplementary figure 2 and copy it as below (P.3 in the revised version with tracked changes):

“Supplementary Figure 2 | Topography of significant tagging response sensors for each participant. A Monte-Carlo based permutation test was performed to select sensors that showed stronger coherence to the tagging signal during the pre-target fixation period compared with the baseline period (no flicker). Significant sensors are marked with circles in the topography, with the values indicating coherence difference (pre-target minus baseline). Participants who had significant tagging response sensors are shown with a bold subtitle. Further coherence analyses (Fig.4, Fig.5, and Supplementary Figure 3) were based on the marked sensors.”

3. Monte-Carlo is the general resampling approach, what FieldTrip does is a permutation – but you need to report what was actually computed: my understanding is that the z-scored mean difference in coherence between baseline and pre-targets is computed, then permuting baseline/pre-target to assess significance (note this is still parametric - the parametric being the mean; non parametric would be rank based) – this should be rephrased accordingly.

We thank the Reviewer for this comment. Now we rephrase the permutation procedure in the new manuscript (P.23 with tracked changes):

“After obtaining the z-scored values for the empirical coherence differences, a permutation procedure was conducted to estimate the statistical significance. We randomly shuffled the trial labels between pre-target and baseline conditions 10,000 times. During each permutation, coherence was computed for both conditions (with shuffled labels), then entered in the above formula to obtain a z score value for the randomization procedure.”

4. Lexical parafoveal previewing facilitates reading: there is no method description here; I imagine this is simply the mean speed per subjects vs mean coherence. To strengthen this claim, could the correlation done per sentence/item? (use a HLM to derive the mean correlation per subject and assess if that mean differs from the null)

We clarify the correlation analysis in the new manuscript as below (P.10 with tracked changes):

“We correlated the pre-target coherence difference (low minus high lexical target frequency) with individual reading speed. Reading speed was quantified as the number of words read per second (i.e. the number of words in all sentences divided by the total reading time).”

We thank the Reviewer for this suggestion. However, in the current analysis, the coherence is estimated over trials, no coherence value for single trial. Potentially we could do a median split in fast versus slowly read sentences; however, we would not be able to then control for confounds such as word length. We also tried to estimate Inter-site phase clustering (ISPC, Lachaux et al, 2000), where phase consistency was obtained over time points (instead of trials) on a single-trial level. But no significant lexical parafoveal previewing effect was observed from a t-test of ISPC. Thus, ISPC turned out to be less sensitive than coherence estimation. We could not find a good estimation for coherence on a single trial level, which makes HLM impossible to conduct.

5. Data availability: given nowadays standards this seems largely inadequate. The material with details of items and matlab PPT script can be made available in any repository; this would also

greatly impact reuse of the RIFT approach. Similarly, csv files for data and R script in plots in fig 2, 4, 5 could be shared, allowing future quantitative comparisons.

Sharing of MEG data can be more difficult and you may have ethical constraints too, all this should be explained in the statement. If shared upon request, which data structure will be used? BIDS? how will you ensure people can reuse the data, is it otherwise documented and already archived. Which conditions apply to sharing? Dr Pan is currently a post-doc, what will happen when/if she moves away – who will be able to provide access to the data?

We thank the Reviewer for this comment about data availability. It's of great importance to share the codes and data in order to facilitate replication of this study. We have now created a repository on OSF (<https://osf.io/ard6h/>) to share the following content:

- Data analysis scripts (.m file), statistic scripts (.R and .m file), and figure plotting scripts (.m file) for Fig. 2, 4, 5, 6 in the main text, and all Supplementary figures.
- Psychtoolbox scripts for the paradigm presentation (.m file)
- Eye-link file for each participant (.edf file).

Due to the storage limit of OSF (50G for public project) and temporary unavailability of the Open MEG Archive (<https://www.mcgill.ca/bic/omega-registration>), the following data are shared upon request: (We will upload them to OMEGA when it is available)

- MEG raw data, epochs, MRI images

We rewrote the Data and code availability section in the new manuscript as below (P.26 with tracked changes):

“Experimental paradigm scripts (Psychtoolbox), statistic scripts (R), scripts to generate all figures, and Eye-link file for each participant are available on OSF (<https://osf.io/ard6h/>). Raw MEG data and T1 images are available from the corresponding author upon request.”

REVIEWER COMMENTS

Reviewer #1 (Remarks to the Author):

I thank the reviewers for addressing my comments. I am happy with the responses.

Reviewer #2 (Remarks to the Author):

The authors have properly addressed our concerns and requests for clarifications. Congratulations to a nice manuscript.

Reviewer #4 (Remarks to the Author):

I have no other comments - this is a great study.
Dr Cyril Pernet